

# Development and Analysis of Soil Water Infiltration Global Database

Mehdi Rahmati[1,2,*], Lutz Weihermüller[2,3], Jan Vanderborght[2,3], Yakov A. Pachepsky[4], Lili Mao[5], Seyed Hamidreza Sadeghi[6], Niloofar Moosavi[2], Hossein Kheirfam[7], Carsten Montzka[2,3], Kris Van Looy[2,3], Brigitta Toth[8], Zeinab Hazbavi[6], Wafa Al Yamani[9], Ammar A. Albalasmeh[10], Ma'in Z. Alghzawi[10], Rafael Angulo-Jaramillo[11], Antônio Celso Dantas Antonino[12], George Arampatzis[13], Robson André Armindo[14], Hossein Asadi[15], Yazidhi Bamutaze[16], Jordi Batlle-Aguilar[17,18,19], Béatrice Béchet[20], Fabian Becker[21], Günter Blöschl[22,23], Klaus Bohne[24], Isabelle Braud[25], Clara Castellano[26], Artemi Cerdà[27], Maha Chalhoub[17], Rogerio Cichota[28], Milena Císlerová[29], Brent Clothier[30], Yves Coquet[17,31], Wim Cornelis[32], Corrado Corradini[33], Artur Paiva Coutinho[12], Muriel Bastista de Oliveira[34], José Ronaldo de Macedo[35], Matheus Fonseca Durães[14], Hojat Emami[36], Iraj Eskandari[37], Asghar Farajnia[38], Alessia Flammini[33], Nándor Fodor[39], Mamoun Gharaibeh[10], Mohamad Hossein Ghavimipanah[6], Teamrat A. Ghezzehei[40], Simone Giertz[41], Evangelos G. Hatzigiannakis[13], Rainer Horn[42], Juan José Jiménez[43], Diederik Jacques[44], Saskia Deborah Keesstra[45,46], Hamid Kelishadi[47], Mahboobeh Kiani-Harchegani[6], Mehdi Kouselou[1], Madan Kumar Jha[48], Laurent Lassabatere[11], Xiaoyan Li[49], Mark A. Liebig[50], Lubomír Lichner[51], María Victoria López[52], Deepesh Machiwal[53], Dirk Mallants[54], Micael Stolben Mallmann[55], Jean Dalmo de Oliveira Marques[56], Miles R Marshall[57], Jan Mertens[58], Félicien Meunier[59], Mohammad Hossein Mohammadi[15], Binayak P Mohanty[60], Mansonia Pulido Moncada[61], Suzana Montenegro[62], Renato Morbidelli[33], David Moret-Fernández[52], Ali Akbar Moosavi[63], Mohammad Reza Mosaddeghi[47], Seyed Bahman Mousavi[1], Hasan Mozaffari[63], Kamal Nabiollahi[64], Mohammad Reza Neyshabouri[65], Marta Vasconcelos Ottoni[66], Theophilo Benedicto Ottoni Filho[67], Mohammad Reza Pahlavan Rad[68], Andreas Panagopoulos[13], Stephan Peth[69], Pierre-Emmanuel Peyneau[20], Tommaso Picciafuoco[22,33], Jean Poesen[70], Manuel Pulido[71], Dalvan José Reinert[72], Sabine Reinsch[57], Meisam Rezaei[32], Francis Parry Roberts[57], David Robinson[57], Jesús Rodrigo-Comino[73,74], Otto Corrêa Rotunno Filho[75], Tadaomi Saito[76], Hideki Suganuma[77], Carla Saltalippi[33], Renáta Sándor[39], Brigitta Schütt[21], Manuel Seeger[74], Nasrollah Sepehrnia[78], Ehsan Sharifi Moghaddam[6], Manoj Shukla[79], Shiraki Shutaro[80], Ricardo Sorando[25], Ajayi Asishana Stanley[81], Peter Strauss[82], Zhongbo Su[83], Ruhollah Taghizadeh-Mehrjardi[84], Encarnación Taguas[85], Wenceslau Geraldes Teixeira[86], Ali Reza Vaezi[87], Mehdi Vafakhah[6], Tomas Vogel[29], Iris Vogeler[28], Jana Votrubova[29], Steffen Werner[88], Thierry Winarski[11], Deniz Yilmaz[89], Michael H. Young[90], Steffen Zacharias[91], Yijian Zeng[83], Ying Zhao[92], Hong Zhao[83], Harry Vereecken[2,3]

1) Department of Soil Science and Engineering, Faculty of Agriculture, University of Maragheh, Maragheh, Iran
2) Forschungszentrum Jülich GmbH, Institute of Bio- and Geosciences: Agrosphere (IBG-3), Jülich, Germany
3) ISMC International Soil Modeling Consortium
4) USDA-ARS Environmental Microbial and Food Safety Laboratory, Beltsville, MD 20705
5) Key Laboratory of Dryland Agriculture, Ministry of Agriculture, Institute of Environment and Sustainable Development in Agriculture, Chinese Academy of Agricultural Sciences, Beijing 100081, PR China
6) Department of Watershed Management Engineering, Faculty of Natural Resources, Tarbiat Modares University, Iran
7) Department of Environmental Sciences, Urmia Lake Research Institute, Urmia University, Urmia, Iran
8) Institute for Soil Sciences and Agricultural Chemistry, Centre for Agricultural Research, Hungarian Academy of Sciences, Budapest, Hungary; University of Pannonia, Georgikon Faculty, Department of Crop Production and Soil Science, Keszthely, Hungary
9) Environment Agency - Abu Dhabi, UAE
10) Department of Natural Resources and Environment, Faculty of Agriculture, Jordan University of Science and Technology, P.O. Box 3030, Irbid 22110, Jordan



11) Univ Lyon, Université Claude Bernard Lyon 1, CNRS, ENTPE, UMR5023 LEHNA, F-69518, Vaulx-en-Velin,
France
12) Universidade Federal de Pernambuco, Centro Acadêmico do Agreste, Núcleo de Tecnologia, Caruaru, Brazil
13) Hellenic Agricultural Organization, Soil and Water Resources Institute, 57400 Sindos, Greece
14) Núcleo de Atividades em Engenharia de Biossistemas (NAEB), DSEA-UFPR, Curitiba, PR, Brazil
15) Department of soil science engineering, Faculty of agriculture and natural resources, University of Tehran. Karaj.
Iran
16) Department of Geography, Geo-Informatics and Climatic Sciences, Makerere University, P.O. Box 7062,
Kampala, Uganda
17) UMR 1402 INRA AgroParisTech Functional Ecology and Ecotoxicology of Agroecosystems, Institut National de
la Recherche Agronomique, AgroParisTech B.P. 01 F-78850 Thiverval-Grignon France
18) UMR 8148 IDES CNRS/Université Paris-Sud XI Bât. 504 Faculté des Sciences 91405 Orsay Cedex, France
19) Innovative Groundwater Solutions (IGS), Victor Harbor, 5211, South Australia, Australia
20) LUNAM Université, IIFSTTAR, GERS, EE, F-44344 Bouguenais, France
21) Freie Universität Berlin, Department of Earth Sciences, Institute of Geographical Sciences, Malteserstr. 74-100,
Lankwitz, 12249, BERLIN, Germany
22) Centre for Water Resource Systems, TU Wien, Karlsplatz 13, 1040 Vienna, Austria
23) Institute of Hydraulic Engineering and Water Resources Management, TU Wien, Karlsplatz 13/222, 1040 Vienna,
Austria
24) Faculty of Agricultural and Environmental Sciences, University of Rostock, Germany
25) Irstea, UE RiverLy, Lyon-Villeurbanne Center, 69625 Villeurbanne, France
26) Pyrenean Institute of Ecology-CSIC. AV. Montañana 1005 // Av. Victoria s / n. 50059 Zaragoza//22700 Jaca,
Huesca. Spain
27) Soil Erosion and Degradation Research Group, Department of Geography, University of Valencia, Valencia, Spain
28) Plant and Food Research, Mount Albert Research Station, Auckland, New Zealand
29) Czech Technical University in Prague, Faculty of Civil Engineering, Thákurova 7, 166 29 Prague 6, Czech
Republic
30) Plant & Food Research, Palmerston North, New Zealand
31) ISTO UMR 7327 Université d'Orléans, CNRS, BRGM, 45071 Orléans, France
32) Department of Soil Management, UNESCO Chair on Eremology, Ghent University, Belgium
33) Department of Civil and Environmental Engineering, University of Perugia, Perugia, Italy
34) UniRedentor University Center. BR 356, 25, Presidente Costa e Silva, Itaperuna, Rio de Janeiro, Brazil
35) Embrapa Solos, Rua Jardim Botânico, 1.024, CEP 22040-060, Jardim Botânico, Rio de Janeiro, RJ, Brazil
36) Department of Soil Science, Faculty of Ferdowsi Mashhad, Mashhad, Iran
37) Dryland Agricultural Research Institute, Agricultural Research, Education and Extension Organization Maragheh,
East Azerbaijan, Iran
38) Scientific Member of Soil and Water Research Department, East Azerbaijan Agricultural and Natural Resources
Research and Education center, Iran
39) Agricultural Institute, Centre for Agricultural Research, Hungarian Academy of Sciences, Brunszvik str. 2., H-
2462 Martonvásár, Hungary
40) Life and Environmental Sciences, University of California, Merced, United States
41) Geographisches Institut, Universität Bonn, Germany
42) Institute of Plant Nutrition and Soil Science, Christian-Albrechts-University zu Kiel, Olshausenstr. 40, 24118 Kiel,
Germany
43) Instituto Pirenaico de Ecología, Spanish National Research Council (IPE-CSIC), Avda. Llano de la victoria 16,
Jaca (Huesca), 22700, Spain
44) Belgian Nuclear Research Centre, Engineered and Geosystems Analysis, Belgium
45) Soil Physics and Land Management Group, Wageningen University, Droevendaalsesteeg 4 6708PB, Wageningen,
The Netherlands
46) Civil, Surveying and Environmental Engineering, The University of Newcastle, Callaghan 2308, Australia
47) Department of Soil Science, College of Agriculture, Isfahan University of Technology, Isfahan 84156-83111, Iran
48) Agricultural & Food Engineering Department, Indian Institute of Technology Kharagpur, Kharagpur – 721302,
West Bengal, India
49) State Key Laboratory of Earth Surface Processes and Resource Ecology, Faculty of Geographical Science, Beijing
Normal University, Beijing 100875 China



50) Research Soil Scientist, USDA Agricultural Research Service, P.O. Box 459, 1701 10th Ave., S.W. Mandan, ND
58554-0459, USA
51) Institute of Hydrology, Slovak Academy of Sciences, Bratislava, Slovakia
52) Departamento de Suelo y Agua, Estación Experimental de Aula Dei (EEAD), Consejo Superior de Investigaciones
Científicas (CSIC), PO Box 13034, 50080 Zaragoza, Spain
53) ICAR-Central Arid Zone Research Institute, Regional Research Station, Kukma – 370105, Bhuj, Gujarat, INDIA
54) CSIRO Land and Water, Glen Osmond, South Australia, Australia
55) PhD student at his first year. Soil Science Graduate Program (ufsm.br/ppgcs), Federal University of Santa Maria,
state of Rio Grande do Sul, Brazil
56) Federal Institute of Education, Science and Technology of the Amazonas – IFAM, Campus Center of Manaus,
Manaus, Brazil
57) Centre for Ecology and Hydrology, Environment Centre Wales, Deiniol Road, Bangor, Gwynedd LL57 2UW,
UK
58) ENGIE Research and Technologies, Simon Bolivardlaan 34, 1000 Brussels, Belgium
59) Universite Catholique de Louvain, Earth and Life Institute-Environmental Sciences, Louvain-la Neuve, Belgium
60) Dep. of Biological and Agricultural Engineering, 2117 TAMU, Texas A&M Univ., College Station, TX 77843-
118    2117
61) Instituto de Edafologı́a, Facultad de Agronomı́a, Universidad Central de Venezuela, Venezuela
62) Universidade Federal de Pernambuco (UFPE), Av. Prof. Moraes Rego, 1235 - Cidade Universitária, Recife - PE
- CEP: 50670-901. Brazil
63) Department of Soil Science, College of Agriculture, Shiraz University, Shiraz, Iran
64) Department of Soil Science and Engineering, Faculty of Agriculture, University of Kurdistan, Sanandaj, Kurdistan
Province, Iran
65) Department of Soil Science, Faculty of Agriculture, University of Tabriz, Tabriz, Iran
66) Department of Hydrology, Geological Survey of Brazil (CPRM), Av. Pauster, 404. CEP 22290-240 Rio de Janeiro
(RJ), Brazil
67) Department of Water Resources and Environment, Federal University of Rio de Janeiro, Avenida Athos da Silveira
Ramos, PO box: 68570, Rio de Janeiro, RJ, Brazil
68) Soil and Water Research Department, Sistan Agricultural and Natural Resources Research and Education Center,
Agricultural Research, Education and Extension Organization (AREEO), Zabol, Iran
69) Department of Soil Science, University of Kassel, Nordbahnhofstr. 1a, 37213 Witzenhausen, Germany
70) Department of Earth and Environmental Sciences, Catholic University of Leuven, Geo-Institute, Celestijnenlaan
200E, 3001 Heverlee, Belgium
71) GeoEnvironmental Research Group, University of Extremadura, Faculty of Philosophy and Letters, Avda. de la
Universidad s/n, 10071 Cáceres, Spain
72) Soil Science Department, Federal University of Santa Maria, state of Rio Grande do Sul, Brazil
73) Instituto de Geomorfología y Suelos, Department of Geography, University of Málaga, 29071, Málaga, Spain
74) Department of Physical Geography, Trier University, D-54286 Trier, Germany
75) Civil Engineering Program, Alberto Luiz Coimbra Institute for Postgraduate Studies and Research in Engineering
(COPPE), Federal University of Rio de Janeiro, Avenida Athos da Silveira Ramos, Rio de Janeiro, RJ, Brazil
76) Faculty of Agriculture, Tottori University, 4-101 Koyama-Minami, Tottori 680-8553, Japan
77) Department of Materials and Life Science, Seikei University, 3-3-1, Kichijoji-kitamachi, Musashino, Tokyo 180-
8633, Japan
78) Department of Soil Science, College of Agriculture, Isfahan University of Technology, Isfahan, Iran
79) Plant and Environmental Sciences, New Mexico State University, Las Cruces, New Mexico
80) Japan International Research Center for Agricultural Science, Rural Development Division, Tsukuba, Japan
81) Department of Agricultural and Bio-Resources Engineerin, Faculty of Engineering, Ahmadu Bello University
Zaria, Nigeria
82) Institute for Land and Water Management Research, Federal Agency for Water Management, Pollnbergstraße 1,
3252 Petzenkirchen, Austria
83) Department of Water Resources, ITC Faculty of Geo-Information Science and Earth Observation, University of
Twente, Enschede, the Netherlands
84) Faculty of Agriculture and Natural Resources, Ardakan University, Ardakan, Yazd Province, Iran
85) University of Córdoba, Department of Rural Engineering, 14071 Córdoba, Spain
86) Soil Physics, Embrapa Soils, Rua Jardim Botaˆnico, 1026, 22460-00 Rio de Janeiro, RJ, Brazil
87) Department of Soil Science, Agriculture Faculty, University of Zanjan, Zanjan, Iran



88) Department of Geography, Ruhr-University Bochum, D-44799 Bochum, Germany
89) Engineering Faculty, Civil Engineering Department, Munzur University, Tunceli, Turkey
90) Bureau of Economic Geology, John A. and Katherine G. Jackson School of Geosciences, University of Texas at
Austin, University Station, Box X, Austin, TX
91) UFZ Helmholtz Centre for Environ. Res., Monitoring and Exploration Technologies, Leipzig, Germany
92) College of Resources and Environmental Engineering, Ludong University, Yantai 264025, China
*) Correspondence to: Mehdi Rahmati (mehdirmti@gmail.com)
**Abstract**
In this paper, we present and analyze a global database of soil infiltration measurements, the Soil Water Infiltration
Global (SWIG) database, for the first time. In total, 5023 infiltration curves were collected across all continents in the
SWIG database. These data were either provided and quality checked by the scientists who performed the experiments
or they were digitized from published articles. Data from 54 different countries were included in the database with
major contributions from Iran, China, and USA. In addition to its global spatial coverage, the collected infiltration
curves cover a time span of research from 1976 to late 2017. Basic information on measurement location and method,
soil properties, and land use were gathered along with the infiltration data, which makes the database valuable for the
development of pedo-transfer functions for estimating soil hydraulic properties, for the evaluation of infiltration
measurement methods, and for developing and validating infiltration models. Soil textural information (clay, silt, and
sand content) is available for 3842 out of 5023 infiltration measurements (~76%) covering nearly all soil USDA
textural classes except for the sandy clay and silt classes. Information on the land use is available for 76 % of
experimental sites with agricultural land use as the dominant type (~40%). We are convinced that the SWIG database
will allow for a better parameterization of the infiltration process in land surface models and for testing infiltration
models. All collected data and related soil characteristics are provided online in *.xlsx and *.csv formats for reference,
and we add a disclaimer that the database is for use by public domain only and can be copied freely by referencing it.
Supplementary data are available at https://doi.pangaea.de/10.1594/PANGAEA.885492. Data quality assessment is
strongly advised prior to any use of this database. Finally, we would like to encourage scientists to extend/update the
SWIG by uploading new data to it.
**Keywords**: Infiltration, Land surface models, Land use, Pedo-transfer functions
**1    Introduction**
Infiltration is the process by which water enters the soil surface and it is one of the key fluxes in the hydrological cycle
and the soil water balance. Water infiltration and the subsequent redistribution of water in the subsurface are two
important processes that affect the soil water balance (Campbell, 1985; Hillel, 2003; Lal and Shukla, 2004; Morbidelli
et al., 2011) and influence several soil processes and functions including availability of water and nutrients for plants,
microbial activity, erosion rates, chemical weathering, and soil thermal and gas exchange between the soil and the
atmosphere (Campbell, 1985). The generation of surface runoff, a key factor in controlling floods, is also directly
related to the infiltration process. Water that cannot infiltrate in the soil becomes available for surface runoff. For these





reasons, infiltration plays a definitive role in maintaining soil system functions and as it is a key process that controls
several of the United Nations Goals for Sustainability (Keesstra et al., 2016).
The infiltration process is usually studied by determining the infiltrated amount of water versus time, from which the
cumulative infiltration, $I$(t), [L], and the infiltration rate, i(t), [L T$^{-1}$] can be derived. i(t) and $I$(t) are related to each
other by derivation (Campbell, 1985; Hillel, 2003; Lal and Shukla, 2004):
$$i\left(t\right)=\frac{dI\left(t\right)}{dt} \tag{1}$$

In general, the soil infiltration rate decreases nonlinearly over time and approaches a constant value after long
infiltration time. Infiltration into the soil is controlled by several factors including soil properties (e.g., texture, bulk
density, initial water content), layering, slope, cover condition (vegetation, crust, and/or stone), rainfall pattern (Smith
et al., 2002; Corradini et al., 2017) and time. As soil texture and soil surface conditions (e.g., cover) are independent
of time at the scale of individual infiltration events, these characteristics can be assumed to be constant during the
event. On the other hand, soil structure, especially at the soil surface, can rapidly change, for instance, due to tillage,
grazing or the destruction of soil aggregates by rain drop impact. In dry soils, initial infiltration rates are substantially
higher than the saturated hydraulic conductivity of the surface layer due to capillary effects which control the sorptivity
of the soil. However, as infiltration proceeds, the gradient between the pressure head at the soil surface and the pressure
head below the wetting front reduces over time so that the infiltration rate finally reaches a constant value that
approximates saturated hydraulic conductivity (Chow et al., 1988).
Infiltration measurements have been largely used to estimate soil saturated hydraulic conductivity. This soil property
is a key to correctly describe all the components of the soil and land surface hydrologic balance and is essential in the
appropriate design of irrigation systems. Large efforts have been invested in literature to estimate this property from
basic soil properties using pedo-transfer functions (PTFs). PTFs are knowledge-based rules or equations that relate
simple soil properties to those properties of soil that are more difficult to obtain (Van Looy et al., 2017). Most of these
efforts have been based on measurements made samples of disturbed or undisturbed soil material. With this infiltration
database, data is now made available that may contribute to better predict the saturated soil hydraulic conductivity and
demonstrate the effect of e.g. vegetation and land management on the parameters of interest.
The Richards (1931), Eq. (2), written as a function of soil water content can be used to derive the closed-form
expression of the infiltration rate in partially saturated soils.
$$\frac{\partial\theta}{\partial t}=\frac{\partial}{\partial z}\left(D_z\left(\theta\right)\frac{\partial\theta}{\partial z}+K_z\left(\theta\right)\right) \tag{2}$$

where $\theta$ is the volumetric soil water content [L$^3$ L$^{-3}$], $t$ is the time [T], $z$ is the vertical depth position [L], $K(\theta)$ is the
soil hydraulic conductivity [L T$^{-1}$], and $D(\theta)$ is soil water diffusivity [L$^2$ T$^{-1}$], which is defined by Eq. (3) (Childs and
Collis-George, 1950; Klute, 1952):
$$D_z\left(\theta\right)=K_z\left(\theta\right)\frac{\partial h}{\partial\theta} \tag{3}$$

where $h$ is the matric potential in head units [L]. The exact relationships between soil water content, soil matric
potential, and soil hydraulic conductivity are necessary to solve the Richards equation. Several solutions of Richards





equation and many empirical/conceptual/semi-analytical/physically-based models, e.g., Green and Ampt (1911);
Philip (1957); Smith and Parlange (1978); Haverkamp et al. (1994); Corradini et al. (2017), have been introduced to
describe the infiltration process over time, even for preferential flows, e.g. Lassabatere et al. (2014). Furthermore,
several direct or indirect experimental systems have been introduced to measure soil infiltration at the laboratory or
in the field under different conditions (Gupta et al., 1994; McKenzie et al., 2002; Mao et al., 2008a). Data obtained
from these systems can also be used to deduce soil saturated hydraulic conductivity directly.
Methods developed to measure and quantify water infiltration in soil are generally time consuming and costly.
Therefore, PTFs have been developed and applied by many researchers, e.g., Jemsi et al. (2013), Parchami-Araghi et
al. (2013), Kashi et al. (2014), Sarmadian and Taghizadeh-Mehrjardi (2014), and Rahmati (2017), in order to easily
parameterize infiltration models. However, these PTFs have been developed for specific regions often limiting their
applicability. As already mentioned, a large number of publications reporting soil infiltration data is available, but
these data are dispersed in the literature and often difficult to access. Therefore, the aim of this data paper is to present
and make available a collection of infiltration data digitized from available literature and from published or
unpublished data provided directly by researchers around the world. These data are accompanied by metadata, which
provide information about the location of infiltration measurement, soil properties, and land management. Finally, we
will provide some first results highlighting the suitability of the database for further research.

## 2  Method and Materials

### 2.1  Data collection

We collected infiltration measurements from all over the globe by contacting the data owners or by extracting
infiltration data from published literature. To do this, a data request was sent to potential data owners through different
forums and email exchanges. The flyer asked data owners to cooperate in the development of the SWIG database by
providing infiltration data as well as metadata about experimental conditions (e.g. initial soil moisture content at the
start of the experiment, method used), soil properties, land use, topography, geographical coordinates of the sites and
any other information relevant to interpret the data and to increase the value of the database. Infiltration data reported
in literature were digitized and included in the database together with additional information provided in these papers.
The digitization approach is discussed in Sect. 2.2. In total, 5023 single infiltration curves were collected of which
510 infiltration curves were digitized from 74 published papers (Table 1) and 4513 were provided by 68 different
research teams (Tables 2 and 3) being published or unpublished data. The references and correspondences for data
supplied by direct communications with researchers are also reported in Tables 2 and 3. Therefore, users may refer to
these references for detailed information about the applied methods or procedures.

<<Table 1 about here>>
<<Table 2 about here>>
<<Table 3 about here>>



## 2.2 Data digitization

In order to digitize infiltration curves reported in literature, screenshots of the relevant plots were taken, and figures were imported into the *plot digitizer 2.6.8* (Huwaldt and Steinhorst, 2015). First, the origin of the axes as well as the highest *x* and *y*-values were defined and the diagram plane was spanned. Then, all point values were picked out and an output table with the $x - y$ pairs (time *vs.* infiltration rate or cumulative infiltration) was generated and stored.

## 2.3 Database structure

The SWIG database is prepared in *.xlsx with a backup file in *.CSV formats containing several datasets. Supplementary data are available at https://doi.pangaea.de/10.1594/PANGAEA.885492. The first dataset, named *I_cm,* contains cumulative infiltration data in centimeter units, and are referred to as *Ixxxx*, whereby *xxxx* is the identifier of the individual infiltration test. The corresponding time intervals in hours for the infiltration data are labeled *T_Hour* and named *Txxxx*. The constant or varying pressure or tension heads (if any) during infiltration measurements are also reported in another dataset named *Tension_cm*. The database also contains additional variables and information relevant to the infiltration data provided by data owners or digitized from articles, as listed in Table 4, and which is labelled *Metadata*. Since the geometric mean diameter ($d_g$) and standard deviation ($S_g$) of soil particle sizes are rarely measured, both parameters were computed using the following equations (Shirazi and Boersma, 1984):

$$d_g = \exp(a), \quad a = 0.01 \sum_{i=1}^{n} f_i \ln M_i \tag{4}$$

$$S_g = \exp(b), \quad b^2 = 0.01 \sum_{i=1}^{n} f_i \ln^2 M_i - a^2 \tag{5}$$

where $f_i$ is the percent of total soil mass having diameters equal to or less than $M_i$, $i$ corresponds to clay, silt, and sand fractions having the arithmetic mean of two consecutive particle size limits of 0.01, 0.026, and 1.025 mm, respectively. For the infiltration data, where the soil texture is unknown, $dg$ and $Sg$ could not be calculated and the data field in the database was left empty. The database also contains the locations of the experimental sites in another dataset named *Locations* that provides the approximate latitude and longitudes in decimal degree (dd.dd) format. Tables 2 and 3 are also provided in the SWIG database in two other worksheets named *Ref. for digitized data* and *Ref. for data provided by owner* for corresponding issues.

<< Table 4 about here>>

# 3 Results and Discussion

## 3.1 Spatial and temporal data coverage

The SWIG database consists of 5023 soil water infiltration measurements spread over nearly all continents (Fig. 1). Data were derived from 54 countries (Table 5). The largest number of data sources were provided by scientists in Iran (n = 38), China (n = 23), and the USA (n = 15), whereby one data source might contain several water infiltration measurements. The SWIG database covers measurements from 1976 to 2017. A low coverage was obtained for the





higher latitudes of the Northern Hemisphere (above 60°) including Norway, Finland, Sweden, Iceland, Greenland,
and Russia. The lack of reports with infiltration data from most countries of the former Soviet Union as well as the
Sahelian and Sahara countries is also notable, as well as the small number of infiltration data from Australia.
Nevertheless, the wide spatial and temporal distribution of infiltration data from this database provides a
comprehensive view on the infiltration characteristics of many soils in the world which can be used in future studies.
<<Figure 1 about here>>
<<Table 5 about here>>
**3.2    Analysis of the database using soil properties**
Textural information (clay, silt, and sand content) are available for 3842 out of 5023 collected infiltration curves (~
76%). The infiltration measurements nearly cover all soil textural classes according to the USDA classification, except
for the sandy clay and silt textural class (Fig. 2), that is of the most important advantages of the SWIG database.
Because soils with extreme textures (clays, very sandy and stony soils) usually are less represented in studies focusing
on their infiltration characteristics (Table 6) as well as their hydrological and erosional response (Poesen, 2018). Loam,
sandy loam, silty loam, and clay loam contributed with 19, 18, 14, and 13 % (Table 6) to the infiltration measurements,
respectively. Table 6 shows that infiltration measurements are almost equally distributed among textures when these
are categorized in three major classes: course- (1092), medium- (1238), and fine to moderately fine-textured soils
(1447). Table 7 reports on the soil properties that are available in SWIG and it gives some simple statistics such as
mean, minimum, maximum, median, and coefficient of variation. Bulk density (available for 66 % of infiltration
measurements) and organic carbon content (available for 62 % of infiltration measurements) are two other soil
properties besides texture that have the highest frequency of availability. Saturated hydraulic conductivity, initial soil
water content, saturated soil water content, calcium carbonate equivalent, electrical conductivity, and pH are available
in 22 to 38 % of infiltration data. The other soil properties have a frequency lower than 10 %. Figure 3 gives a general
overview of cumulative infiltration curves for the different soil textural groups listed in Table 6.
<<Figure 2 about here>>
<<Table 6 about here>>
<<Table 7 about here>>
<<Figure 3 about here>>
**3.3    Infiltration measurements in the SWIG database**
Different instruments were used to measure soil water infiltration (Table 8). About 32% (1595 out of 5023) of the
measurements were carried out using different types of ring infiltrometers. The most frequently used methods are the
disc infiltrometer methods (disc, mini-disc, and micro-disc, hood, and tension infiltrometers), which have been used
in about 51% of the experiments. About 5% of the data were submitted to the database without specifying the
measurement method (251 infiltration tests) and around 12 % of the measurements were carried out with other methods
not listed above (Table 8).
<<Table 8 about here>>



### 3.4    Land use classes represented in the SWIG database

Since land use is one of the most important factors affecting soil surface processes including water infiltration in soils, we collected information on the type of land use at all the experimental sites when available. In general, the type of land use was reported in 3818 out of 5023 infiltration curves (~76 %) and information is reported in the *Metadata* dataset. For simplicity, we grouped all reported land use types into 22 major groups (Table 9). A frequency analysis showed that agricultural land use, i.e. cropped land, irrigated land, dryland, and fallow land, is the most frequently reported land use in the database with about 53% (2019 out of 3818) of all land uses. Grassland represents with 22% the second largest land use type. Pasture with 6 % and forest with 5 % are ranked as third and fourth largest reported land use types. The 18 remaining land use types all together cover only 545 experimental sites (less than 15%). The cumulative infiltration curves for four dominant land-use types are shown in Fig. 4 in order to give a general overview on the magnitudes and spread of cumulative infiltration between the different land uses. It is striking that all four land uses show upper and lower cumulative infiltration values that are very similar.

<<Table 9 about here>>

<<Figure 4 about here>>

### 3.5    Estimating infiltration parameters from infiltration measurements

In order to predict infiltration parameters from infiltration measurements, we classified the SWIG infiltration curves in two groups: i) infiltration curves that were obtained under the assumption of 1D infiltration and ii) infiltration curves that were obtained under 3D flow conditions. We fitted the three-parameter infiltration equation of Philip (Kutílek and Krejča, 1987), Eq. (6), to the 1D experimental data and the simplified form of Haverkamp et al. (1994), Eq. (7), to the 3D experimental data:

$$I_{1D} = St^{\frac{1}{2}} + A_1 t + A_2 t^{\frac{3}{2}} \tag{6}$$

$$I_{3D} = S\sqrt{t} + \left[ \frac{2-\beta}{3} K_{sat} + \frac{\gamma S^2}{R_D (\theta_s - \theta_i)} \right] t \tag{7}$$

We reduced the number of parameters in Eq. (6) by defining $A_1 = 0.33 \times K_{sat}$ (Philip, 1957) and $A_2 = A$ where $A$ was assumed to be a lumped parameter. In Eq. (7), we put $\beta = 0.6$ (Angulo-Jaramillo et al., 2000) and the second term between brackets on the right hand side was assumed to be a lumped parameter. Therefore, we simplified the equations as follow:

$$I_{1D} = St^{\frac{1}{2}} + 0.33 K_{sat} t + A t^{\frac{3}{2}} \tag{8}$$

$$I_{3D} = S\sqrt{t} + 0.47 K_{sat} t + A t \tag{9}$$

In our analysis, we assumed that double ring infiltrometer measurements result in 1D infiltration conditions, while the different types of disc infiltration and single ring infiltrometer measurements lead to 3D flow conditions that can be captured by Eq. (9). As this is not guaranteed for measurements made with rainfall simulator, Guelph permeameter,





Aardvark permeameter, linear and point source methods as well as Hood infiltrometer measurements, these infiltration
curves were not considered in our first analysis. By excluding these methods, 596 infiltration curves were rejected
from analysis. In addition, 251 infiltration curves were also excluded from analysis as no indication was available on
the measurement method used. In total, 4178 infiltration curves were included in our analysis of which 828 infiltration
curves reflected 1D and 3350 were considered as the results of 3D infiltration. As no sufficient information was
available on the properties of the sand contact layer, we did not correct 3D infiltration measurements. Finally, the
selected infiltration curves were fitted to Eq. (8) or (9) using lsqnonlin command in matlab.
The fitting results of Eq. (8) to the single infiltrometer data are shown in Table 10. $R^2$ values were higher than 0.9 in
97 % of the cases and higher than 0.99 in 77 % of the cases. Fitting Eq. (9) to the 3D infiltration curves data, $R^2$ values
higher than 0.9 for 94 % of the infiltration curves and higher than 0.99 for 68 % of the infiltration curves were obtained.
The statistics for the fitting process as well as the fitted parameters of two mentioned models are reported in the SWIG
database in an additional dataset labelled *Statistics*. For infiltration curves excluded from analysis, an empty cell is
reported.

<<Table 10 about here>>

The average values of estimated $K_{sat}$ and sorptivity ($S$), using Eq. (8) or (9) as well as measured $K_{sat}$ for different soil
texture classes extracted from the current database is reported in Table 11. Comparison between estimated ( $K_{sat-es}$ )
and measured ( $K_{sat-m}$ ) values of $K_{sat}$ (Table 11) reveals that there is reasonably good agreement between
measurements and estimation, except for loamy sand (with mean $K_{sat-es}$ = 62 cm h$^{-1}$ *vs.* $K_{sat-m}$ =25 cm h$^{-1}$), sandy
loam (with mean $K_{sat-es}$ = 32 cm h$^{-1}$ *vs.* $K_{sat-m}$ =41 cm h$^{-1}$), silt loam (with mean $K_{sat-es}$ = 27 cm h$^{-1}$ *vs.* $K_{sat-m}$
=3 cm h$^{-1}$), and silty clay (with mean $K_{sat-es}$ = 26 cm h$^{-1}$ *vs.* $K_{sat-m}$ =45 cm h$^{-1}$) textural classes. However, the only
significant difference between measured and estimated $K_{sat}$ values was found for the silt loam texture class (Table 11)
applying an independent T test.
We also compared our estimated $K_{sat}$ values from the infiltration measurements in SWIG database with $K_{sat}$ values
from databases that have been published in the literature (Table 12). Some of these databases like the one of Clapp
and Hornberger (1978) and Cosby et al. (1984) have been used to parameterize land surface models. Most of the $K_{sat}$
values in the listed databases have been obtained from lab scale measurements often performed on disturbed soil
samples. In most of the reported databases $K_{sat}$ is controlled by texture with the highest mean values obtained for the
coarse textured and the lowest mean values for the fine textured soils. This is not the case for the $K_{sat}$ values obtained
from the SWIG database. Clayey soils have a mean value that is similar to the coarser textured soils. This may be
partly explained by the fact that the measurements collected in the SWIG database are obtained from field
measurements on undisturbed soils. It is also striking that the standard deviation of $K_{sat}$ in the SWIG database is
typically larger than the standard deviations obtained from the databases in literature. This indicates that texture is
apparently not the most important control on $K_{sat}$ values. This finding indicates that present parameterization in
currently used land surface models, which are mainly based on texture, may severely underestimate the variability of
$K_{sat}$. In addition, it shows that also mean values are not dominantly controlled by textural properties. Other land surface
properties such as land use, crusting, etc. may turn out to be much more important.




<<Table 11 about here>>
<<Table 12 about here>>

### 3.6    Exploring the SWIG database using principal component analysis

In order to demonstrate the potential of the SWIG database for analyzing infiltration data and for developing pedo-
transfer functions, principal component analysis (PCA) were performed and biplots were generated to show both the
observations and the original variables in the principal component space (Gabriel, 1971).
In a biplot, positively correlated variables are closely aligned with each other and the larger the arrows the stronger
the correlation. Arrows that are aligned in opposite direction are negatively correlated with each other and the
magnitude of the arrows is again a measure for the strength of the correlation. Arrows that are aligned 90 degrees to
each other show typically no correlation.  Fig. 5 and 6 show the results of two PCA. The first PCA (Fig. 5) shows the
relationship between soil textural properties, $S$ and $K_{sat}$ based on 3267 infiltration measurements. The first two
principal components explain 74.5% of the variability in the data. Figure 5 shows a positive correlation between $K_{sat}$
and $S$ (0.527) and the largest values for both variables are found in clay soils. Clay content appears only to be weakly
correlated with $K_{sat}$ and $S$ as is also shown by correlation coefficients of 0.112 and 0.025 respectively. Figure 6 shows
the biplot of soil textural properties, $K_{sat}$, $S$, organic carbon content, and bulk density in the principal component space
based on 1910 infiltration measurements. The first two principal components still explain 55% of the variability.
Neither $S$ nor $K_{sat}$ showed appreciable correlations with available soil properties. Only $K_{sat}$ and $S$ are correlated (arrows
are aligned but small) with a value of 0.29. Organic carbon and bulk density show a negative correlation with a
calculated value equal to -0.51. It also shows that for example the sandy clay loam textural class (yellow dots) shows
a wide spread in organic matter content and bulk densities. These analyses show that basic soil properties do not
contain enough information to properly estimate $K_{sat}$ and $S$. However, the SWIG database provides additional
information like land use, initial water content and slope that might prove to be good predictors. A further analysis in
this respect is however beyond the scope of this paper. More importantly, the present analysis in combination with the
results provided in Table 12 shows that a texture dominated derivation of $K_{sat}$ values, as done in most land surface
models, does not provide an adequate way to estimate $K_{sat}$.
<<Figure 5 about here>>
<<Figure 6 about here>>

### 3.7    Potential error and uncertainty in the SWIG database

Similar to any other database, the data presented in the SWIG database may be subject to different error sources and
uncertainties. These include: 1) transcription errors that occurred when implementing the measurement data into the
EXCEL spreadsheets, 2) inaccuracy and uncertainties in determining related soil properties such as textural properties,
3) violation of the underlying assumption when performing the experiments, and 4) uncertainty (variability) in
estimated soil hydraulic properties due to the different measurement methods. Unfortunately, none of these error or
uncertainty sources are under the control of the SWIG database authors and quantification of these sources is often
difficult as the required information is often lacking. The uncertainty with respect to the effect of the measurement



techniques on estimated soil hydraulic properties may be quantified as information on the measurement is available.
Yet some of these methods may only have been used in few cases making a statistical analysis difficult.
With respect to the transcription error, intensive attempts have been made to double check data transcription to prevent
or at least to minimize any probable error for this part. Values of soil properties such as textural composition are
known to vary strongly between different labs and measurement methods. This is especially true for the finer textural
classes like clay. Unfortunately, information on the measurement used to determine soil properties is most of the time
lacking or insufficient to assess the magnitude of errors or biases.
The uncertainty with respect to the effect of measurement techniques on quantifying the infiltration process may be
analyzed from the SWIG database as it provides information on the type of measurement technique used. This analysis
is however beyond the scope of the paper. Potential error and uncertainty sources with respect to the use of different
measurements are discussed in the supplementary material.
The uncertainty on estimated soil hydraulic properties from infiltration measurements may be strongly controlled by
the person performing the experiment but may also be due the different measurement windows of the methods in terms
of measurement volume. The SWIG database provides information to quantify uncertainties introduced by difference
in measurement volume and this analysis will be closely related to the assessment of the representative elementary
volume, REV (see e.g. the work of Pachepsky on scaling of saturated hydraulic conductivity).
Another case in the SWIG database that users may find odd is that some water repellent soils, for example the soils
coded 1211 to 1420 in SWIG with very high sand content (>95%), can show relatively low infiltration rates, which
would refer to clay texture rather than sand. However, one may consider that it is a natural phenomenon and not caused
by measuring failure.
One needs to carefully by interpreting estimated of $K_{sat}$ from clayey soils showing high values of $K_{sat}$ (for example the
soils coded 3746 to 3833 in SWIG). The $K_{sat}$ values for these soils were obtained using the single ring infiltrometer
method. These infiltration experiments were conducted in the field under ponded conditions, and with a minimum
disturbance of the natural surface (vegetation was only cut but roots let in place) and evidenced an impact of land use
on $K_{sat}$, that is much higher than the impact of soil texture. Under ponding conditions, macropores can be activated,
and this is all the more likely as a quite large cylinder diameter of 40 cm was used. Very high values were obtained
for forested land uses, and sometimes for grassland, but cracks were present.
**3.8    Research potentials of the SWIG database**
We envision that SWIG offers a unique opportunity and information source to 1) evaluate infiltration methods and to
assess their value in deriving soil hydraulic properties, 2) test different models and concepts for point scale and grid
scale infiltration processes, 3) develop pedo-transfer functions (PTFs) to estimate soil hydraulic properties such as the
Mualem van Genuchten parameters, 4) identify controls on infiltration processes, 5) validate global predictions of
infiltration from land surface models, 6) study more complex processes like preferential flow in soils, and 7) highlight
the state of the art on understanding the relationships between infiltration and several soil surface characteristics, for
example the SWIG database effectively can contribute to the scope of Morbidelli et al. (2018) to advance the
knowledge of infiltration over sloping surfaces.



We are confident that the SWIG database is just a first step in collecting and archiving infiltration data and we expect
that more and more data will become available in the near future. These data will be archived in SWIG and thus made
available to the world-wide research community. In this regard, we are interested in receiving existing or newly
measured infiltration curves and for this purpose the corresponding author will serve as point of contact or data can
be made available through the International Soil Modeling Consortium, ISMC (https://soil-modeling.org/), for further
archiving in SWIG.

## 4  Conclusion

We have collected 5023 infiltration curves from field experiments from all over the world covering a broad range of
soils, land uses and climate regions. We estimated saturated hydraulic conductivity, $K_{sat}$, and sorptivity from more
than 3000 infiltration curves and compared estimated $K_{sat}$ values with values from different databases published in
literature. We showed that contrary to the assumption made in many land surface and global climate models, that
texture is not the controlling factor for $K_{sat}$. In addition, the variability in $K_{sat}$ derived from these field measurements
is considerably larger than reported in literature. The collected infiltration curves were archived as SWIG database on
the PANGAEA platform and are therefore world-wide available. The data are structured into *.xlsx and *.csv files
and include metadata information for further use. Data analysis revealed that infiltration curves are lacking for clayey,
sandy textured and stony soils. Also infiltration curve data are lacking for the Northern and permafrost regions. Here
additional efforts are needed to collect additional data as these regions are sensitive to climate change which will
clearly affect the soil hydrology.

## Acknowledgments

Authors gratefully thank the International Soil Modeling Consortium (ISMC) and the International Soil and Tillage
Organization (ISTRO) for their help in distributing our call for data among researchers in the world. Parts of data were
gathered from work that was supported by the UK-China Virtual Joint Centre for Agricultural Nitrogen (CINAg,
BB/N013468/1), which is jointly supported by the Newton Fund, via UK BBSRC and NERC. The French Claduègne
and Yzeron data sets were acquired during the ANR projects FloodScale (ANR-2011-BS56-027) and AVuUR (ANR-
07-VULN-01) respectively. Also, parts of the database were made available through research work carried out in the
framework of LIFE+ projects funded by the EC. The Spanish Ministry of Economy is acknowledged for support
through project CGL2014-53017-C2-1-R. The Czech Science Foundation is acknowledged for support through project
No. 16-05665S.

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






Table 1- References used to extract infiltration curves and metadata

| No. | Dataset From | Dataset To | Reference | No. | Dataset From | Dataset To | Reference | No. | dataset From | dataset To | Reference |
|---|---|---|---|---|---|---|---|---|---|---|---|
| 1 | 295 | 317 | Miller et al. (2005) | 26 | 4516 | - | Delage et al. (2016) | 51 | 4692 | - | Ayu et al. (2013) |
| 2 | 318 | 322 | Adindu Ruth et al. (2014) | 27 | 4517 | 4518 | Ruprecht and Schofield (1993) | 52 | 4693 | 4699 | Rei et al. (2016) |
| 3 | 542 | 544 | Alagna et al. (2016) | 28 | 4519 | 4520 | Bertol et al. (2015) | 53 | 4700 | 4702 | Omuto et al. (2006) |
| 4 | 545 | - | Angulo-Jaramillo et al. (2000) | 29 | 4521 | 4523 | Naeth et al. (1991) | 54 | 4703 | 4706 | Návar and Synnott (2000) |
| 5 | 546 | 548 | Su et al. (2016) | 30 | 4524 | 4529 | Huang et al. (2011) | 55 | 4707 | - | Scotter et al. (1988) |
| 6 | 549 | 550 | Quadri et al. (1994) | 31 | 4530 | 4537 | van der Kamp et al. (2003) | 56 | 4708 | 4720 | Khan and Strosser (1998) |
| 7 | 551 | 553 | Qi and Liu (2014) | 32 | 4538 | - | Jačka et al. (2016) | 57 | 4721 | 4724 | Lipiec et al. (2006) |
| 8 | 554 | 558 | Huang et al. (2015) | 33 | 4539 | 4568 | Matula (2003) | 58 | 4725 | - | Suzuki (2013) |
| 9 | 559 | 568 | Al-Kayssi and Mustafa (2016) | 34 | 4569 | 4586 | Casanova (1998) | 59 | 4726 | 4728 | Sukhanovskij et al. (2015) |
| 10 | 1421 | 1432 | Bhardwaj and Singh (1992) | 35 | 4587 | 4593 | Holzapfel et al. (1988) | 60 | 4729 | 4749 | Al-Ghazal (2002) |
| 11 | 1433 | 1435 | Berglund et al. (1980) | 36 | 4594 | 4605 | Wang et al. (2015b) | 61 | 4750 | - | Sorman et al. (1995) |
| 12 | 1436 | 1443 | Wu et al. (2016) | 37 | 4606 | 4611 | Mao et al. (2016) | 62 | 4751 | 4764 | Bowyer-Bower (1993) |
| 13 | 1444 | 1446 | Chartier et al. (2011) | 38 | 4612 | - | Wang et al. (2016) | 63 | 4765 | 4788 | Medinski et al. (2009) |
| 14 | 1447 | 1456 | Sihag et al. (2017) | 39 | 4613 | 4615 | Qian et al. (2014) | 64 | 4789 | 4792 | Latorre et al. (2015) |
| 15 | 1457 | 1460 | Machiwal et al. (2006) | 40 | 4617 | 4619 | Fan et al. (2013) | 65 | 4793 | 4795 | Biro et al. (2010) |
| 16 | 1461 | 1466 | Igbadun et al. (2016) | 41 | 4620 | - | Zhang et al. (2000) | 66 | 4796 | 4799 | Mohammed et al. (2007) |
| 17 | 1467 | 1469 | Mohanty et al. (1994) | 42 | 4621 | 4623 | Wang et al. (2015a) | 67 | 4800 | 4815 | Abdallah et al. (2016) |
| 18 | 1470 | 1472 | Sauwa et al. (2013) | 43 | 4624 | 4633 | Yang and Zhang (2011) | 68 | 4816 | 4819 | Murray and Buttle (2005) |
| 19 | 1473 | 1476 | Arshad et al. (2015) | 44 | 4634 | 4657 | Wu et al. (2016) | 69 | 4820 | 4831 | Zhang et al. (2015) |
| 20 | 1477 | 1488 | Bhawan (1997) | 45 | 4658 | 4663 | Ma et al. (2017) | 70 | 4832 | 4837 | Perkins and McDaniel (2005) |
| 21 | 1489 | 1495 | Uloma et al. (2013) | 46 | 4664 | 4681 | Thierfelder et al. (2003) | 71 | 4838 | 4841 | Arriaga et al. (2010) |
| 22 | 1496 | - | Al-Azawi (1985) | 47 | 4682 | 4683 | Commandeur et al. (1994) | 72 | 4842 | 4857 | Thierfelder et al. (2017) |
| 23 | 1497 | 1499 | Ogbe et al. (2011) | 48 | 4684 | 4686 | Di Prima et al. (2016) | 73 | 4858 | 4867 | Thierfelder and Wall (2009) |
| 24 | 1500 | 1507 | Teague (2010) | 49 | 4687 | 4688 | Angulo-Jaramillo et al. (2000) | 74 | 4868 | 4879 | Abagale et al. (2012) |
| 25 | 4506 | 4515 | Muhamad et al. (2008) | 50 | 4689 | 4691 | Machiwal et al. (2006) | | | | |






Table 2- References and correspondence for data supplied by data owners

| No. | Dataset From | Dataset To | Contact person | Email for contact | Reference |
|---|---|---|---|---|---|
| 1 | 1 | 135 | M. Rahmati | mehdirmti@gmail.com | Rahmati (2017) |
| 2 | 136 | 294 | A. Farajnia | farajnia1966@yahoo.com | Unpublished data |
| 3 | 323 | 376 | M. Shukla | shuklamk@nmsu.edu | Shukla et al. (2003 & 2006) |
| 4 | 377 | 426 | S. H. R. Sadeghi | sadeghi@modares.ac.ir | Sadeghi et al. (2014, 2016a, b, c, 2017a, b), Hazbavi and Sadeghi (2016), Kheirfam et al. (2017a, b) Sharifi Moghaddam et al. (2014); Ghavimi Panah et al. (2017); Kiani-Harchegani et al. (2017) |
| 5 | 427 | 466 | M. H. Mohammadi | mhmohmad@ut.ac.ir | Unpublished data |
| 6 | 467 | 505 | F. Meunier | felicien.meunier@uclouvain.be | Unpublished data |
| 7 | 506 | 541 | N. Sephrnia | n.sepehrnia@gmail.com | Sepehrnia et al. (2016 & 2017) |
| 8 | 569 | 817 | D.Moret-Fernández | david@eead.csic.es | Unpublished data |
| 9 | 818 | 940 | M. Vafakhah | vafakhah@modares.ac.ir | Kavousi et al. (2013); Fakher Nikche et al. (2014) |
| 10 | 941 | 1060 | A. Cerdà | artemio.cerda@uv.es | Unpublished data |
| 11 | 1061 | 1079 | J. Rodrigo-Comino | rodrigo-comino@uma.es | Rodrigo-Comino et al. (2016); Rodrigo-Comino et al. (2018) |
| 12 | 1080 | 1112 | H. Asadi | hossein_asadi52@yahoo.com | Nikghalpour et al. (2016) |
| 13 | 1113 | 1119 | K. Bohne | klaus.bohne@uni-rostock.de | Unpublished data |
| 14 | 1120 | 1125 | L. Mao | leoam@126.com | Mao et al. (2008b; 2016) |
| 15 | 1126 | 1166 | L. Lichner | lichner@uh.savba.sk | Dušek et al. (2013), Lichner et al. (2011; 2012; 2013) |
| 16 | 1167 | 1210 | M. V. Ottoni | marta.ottoni@cprm.gov.br | Oliveira (2005) |
| 17 | 1211 | 1420 | R. Sándor | sandor.rencsi@gmail.com | Fodor et al. (2011); Sándor et al. (2015) |
| 18 | 4476 | 4485 | | | |
| 19 | 1508 | 1519 | A. Stanley | ajayistan@gmail.com | Igbadun et al. (2016); Othman and Ajayi (2016) |
| 20 | 1520 | 1521 | A. R. Vaezi | vaezi.alireza@gmail.com | Unpublished data |
| 21 | 1522 | 1536 | A. Albalasmeh | aalbalasmeh@just.edu.jo | Gharaibeh et al. (2016) |
| 22 | 1537 | 1578 | D. Machiwal | dmachiwal@rediffmail.com | Machiwal et al. (2006, 2017), Ojha et al. (2013) |
| 23 | 1579 | 1592 | H. Emami | hemami@um.ac.ir | Fakouri et al. (2011a, 2011b) |
| 24 | 1593 | 1895 | J. Mertens | jan.mertens@engie.com | Mertens et al. (2002, 2004, 2005) |
| 25 | 1896 | 2115 | D. Jacques | diederik.jacques@sckcen.be | Jacques (2000); Jacques et al. (2002) |
| 26 | 2116 | 2139 | J. Votrubova | jana.votrubova@fsv.cvut.cz | Votrubova et al. (2017) |
| 27 | 2140 | 2143 | J. Batlle-Aguilar | jorbat1977@hotmail.com | Batlle-Aguilar et al. (2009) |
| 28 | 2144 | 2179 | R. A. Armindo | rarmindo@ufpr.br | Unpublished data |
| 29 | 2180 | 2209 | S. Werner | steffen.werner@rub.de | Unpublished data |
| 30 | 2210 | 2255 | S. Zacharias | steffen.zacharias@ufz.de | Unpublished data |
| 31 | 2256 | 2281 | S. Shutaro | sshiraki@affrc.go.jp | Unpublished data |
| 32 | 2282 | 2304 | T. Saito | tadaomi@muses.tottori-u.ac.jp | Saito et al. (2016) |
| 33 | 2305 | 2354 | R. Taghizadeh-M. | rh_taghizade@yahoo.com | Unpublished data |

Table 3- References and correspondence for data supplied by data owners (continued by Table 2)

| No. | Dataset From | Dataset To | Contact person | Email for contact | Reference |
|---|---|---|---|---|---|
| 34 | 2355 | 2356 | W. G. Teixeira | wenceslau.teixeira@embrapa.br | Teixeira et al. (2014) |
| 35 | 3644 | 3647 | | | |
| 36 | 2357 | 2436 | Y. Zhao | yzhaosoils@gmail.com | Zhao et al. (2011) |
| 37 | 2437 | 2475 | A. A. Moosavi | aamousavi@gmail.com | Unpublished data |
| 38 | 2476 | 2552 | Y. A. Pachepsky | Yakov.Pachepsky@ars.usda.gov | Rawls et al. (1976) |
| 39 | 2553 | 2643 | A. Panagopoulos | panagopoulosa@gmail.com | Hatzigiannakis and Panoras (2011) + unpublished data |
| 40 | 2644 | 2649 | B. Clothier | Brent.Clothier@plantandfood.co.nz | Al Yamani et al. (2016) |
| 41 | 2650 | 2710 | C. Castellano | ccastellanonavarro@gmail.com | Unpublished data |
| 42 | 3507 | 3597 | | | |
| 43 | 2711 | 2756 | F. Becker | fabian.becker@fu-berlin.de | Unpublished data |
| 44 | 2757 | 2765 | I. Vogeler | iris.vogeler@plantandfood.co.nz | Vogeler et al. (2006); Cichota et al. (2013) |
| 45 | 2766 | 2788 | R. Morbidelli | renato.morbidelli@unipg.it | Morbidelli et al. (2017) |
| 46 | 2789 | 2832 | S. Giertz | sgiertz@uni-bonn.de | Giertz et al. (2005) |
| 47 | 2833 | 2868 | T. Vogel | vogel@fsv.cvut.cz | Vogel and Cislerova (1993) |
| 48 | 2869 | 2948 | W. Cornelis | Wim.Cornelis@ugent.be | Pulido Moncada et al. (2014) |
| 49 | 2949 | 3386 | Y. Coquet | yves.coquet@univ-orleans.fr | Coquet (1996); Coquet et al. (2005); Chalhoub et al. (2009) |
| 50 | 3705 | 3709 | | | |
| 51 | 3387 | 3506 | B. Mohanty | bmohanty@tamu.edu | Das Gupta et al. (2006) |
| 52 | 3598 | 3643 | D. J. Reinert | dalvan@ufsm.br | Mallmann (2017) |
| 53 | 3648 | 3657 | M.R. Pahlavan Rad | pahlavanrad@gmail.com | Pahlavan-Rad (2016) |
| 54 | 3658 | 3680 | T. Saito | tadaomi@muses.tottori-u.ac.jp | Unpublished data |
| 55 | 3681 | 3704 | X. Li | xyli@bnu.edu.cn | Li et al. (2013); Hu et al. (2016) |
| 56 | 4497 | 4505 | | | |
| 57 | 3710 | 3745 | Y. Bamutaze | yazidhibamutaze@gmail.com | Unpublished data |
| 58 | 3746 | 3833 | I. Braud | isabelle.braud@irstea.fr | Gonzalez-Sosa et al. (2010); Braud (2015); Braud and Vandervaere (2015) |
| 59 | 3907 | 4011 | | | |
| 60 | 3834 | 3874 | M. R. Mosaddeghi | mosaddeghi@yahoo.com | Unpublished data |
| 61 | 3875 | 3906 | S. B. Mousavi | b_mosavi2000@yahoo.com | Unpublished data |
| 62 | 4012 | 4026 | M. Pulido | manpufer@hotmail.com | Unpublished data |
| 63 | 4027 | 4457 | F. P. Roberts | frapar@ceh.ac.uk | Unpublished data |
| | 4458 | 4475 | | | Robinson et al. (2016, 2017) |
| 64 | 4486 | 4496 | T. Picciafuoco | picciafuoco@hydro.tuwien.ac.at | Morbidelli et al. (2017) |
| 65 | 4880 | 4886 | M. A. Liebig | mark.liebig@ars.usda.gov | Liebig et al. (2004) |
| 66 | 4887 | 4936 | Y. Zeng | y.zeng@utwente.nl | Zhao et al. (2017, 2018) |
| 67 | 4937 | 5018 | L. Lassabatere | laurent.lassabatere@entpe.fr | Lassabatere et al. (2010); Yilmaz et al. (2010); Coutinho et al. (2016) |
| 68 | 5019 | 5023 | I. Eskandari | eskandari1343@yahoo.com | Unpublished data |







Table 4- Description of the variables listed in database

| Column | Supplies: | Dimension |
|---|---|---|
| *Code* | Data set identifier with 4 digits from 0001 to 5023 | |
| *Clay* | Mass of soil particles, < 0.002 mm | % |
| *Silt* | Mass of soil particles, >0.002 and < 0.05 mm | % |
| *Sand* | Mass of soil particle, > 0.05 and < 2 mm | % |
| *Texture* | 1: Sand; 2: Loamy sand; 3: Sandy loam; 4: Sandy clay loam; 5: Sandy Clay; 6: Loam; 7: Silt loam; 8: Silt; 9: Clay loam; 10: Silty clay loam; 11: Silty clay; 12: Clay | |
| *Gravel* | Mass of particles larger than 2 mm | % |
| *dg* | Geometric mean diameter | mm |
| *Sg* | Standard deviation of soil particle diameter | |
| *OC* | Soil organic carbon content | % |
| *Db* | Soil bulk density | g cm$^{-3}$ |
| *Dp* | Soil particle density | g cm$^{-3}$ |
| *Ksat* | Soil saturated hydraulic conductivity | cm h$^{-1}$ |
| *Theta_sat* | Saturated volumetric soil water content | cm$^3$ cm$^{-3}$ |
| *Theta_i* | Initial volumetric soil water content | cm$^3$ cm$^{-3}$ |
| *FC* | Soil water content at field capacity | cm$^3$ cm$^{-3}$ |
| *PWP* | Soil water content at permanent wilting point (1500 kPa) | cm$^3$ cm$^{-3}$ |
| *Theta_r* | Residual volumetric soil water content | cm$^3$ cm$^{-3}$ |
| *WAS* | Wet-aggregate stability | % |
| *MWD* | Aggregates mean weight diameter | mm |
| *GMD* | Aggregates geometric mean diameter | mm |
| *EC* | Soil electrical conductivity | dS m$^{-1}$ |
| *pH* | Soil acidity | - |
| *Gypsum* | Soil gypsum content | % |
| *CCE* | Soil carbonate calcium equivalent | % |
| *CEC* | Soil cation exchange capacity | Cmol$_c$ kg$^{-1}$ |
| *SAR* | Soil sodium adsorption ratio | - |
| *DiscRadius* | Applied disc radius (if any) | mm |
| *Instrument* | Applied instruments for infiltration measurement: 1: Double ring; 2: Single ring; 3: Rainfall simulator; 4: Guelph permeameter; 5: Disc infiltrometer; 6: Micro-infiltrometer; 7: Mini-infiltrometer; 8: Aardvark Permeameter; 9: Linear source method; 10: Point source method; 11: Hood infiltrometer; 12: Tension infiltrometer; 13: BEST method | |
| *Vegetation cover* | | % |
| *Land use* | Dominant land-use or land cover type of the experimental site | |
| *Rainfall intensity* | Simulated rain intensity | mm h$^{-1}$ |
| *Slope* | The mean slope of the soil surface | % |
| *Treatment* | Applied treatment in experimental site | |
| *Crust* | Yes: existence of crust; No: no crust layer | |
| *Sand contact layer* | Yes: sand contact layer is applied during infiltration measurement; No: no sand contact layer | |


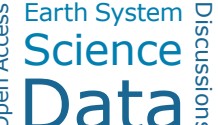




Table 5- Countries and the number of data sources (n) contributing to the database

| Country | n | Country | n | Country | n |
|---|---|---|---|---|---|
| Iran | 38 | Austria | 2 | Indonesia | 1 |
| China | 23 | Chile | 2 | Iraq | 1 |
| USA | 15 | Ghana | 2 | Japan | 1 |
| Brazil | 9 | Morocco | 2 | Jordan | 1 |
| Spain | 9 | Namibia | 2 | Kenya | 1 |
| France | 9 | New Zealand | 2 | Lebanon | 1 |
| Germany | 8 | Pakistan | 2 | Malawi | 1 |
| India | 8 | Russia | 2 | Mexico | 1 |
| Canada | 7 | Senegal | 2 | Mozambique | 1 |
| United Kingdom | 7 | Slovakia | 2 | Myanmar | 1 |
| Hungary | 6 | South Africa | 2 | Netherland | 1 |
| Nigeria | 6 | Sudan | 2 | Poland | 1 |
| Greece | 5 | Zambia | 2 | Scotland | 1 |
| Belgium | 4 | Argentina | 1 | Tanzania | 1 |
| Italy | 4 | Australia | 1 | Telangana | 1 |
| Czech Republic | 3 | Benin | 1 | UAE | 1 |
| Saudi Arabia | 3 | Cameroon | 1 | Uganda | 1 |
| Australia | 2 | Colombia | 1 | Zimbabwe | 1 |


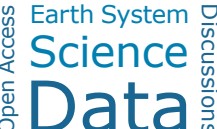

Table 6- Number of soils in each soil USDA textural class for which infiltration data are included in the database.

| Group | Soil texture class | Availability |
|---|---|---|
| Coarse-textured soils | | 1092 |
| | Sand | 291 |
| | Loamy sand | 111 |
| | Sandy loam | 690 |
| Medium-textured soils | | 1238 |
| | Loam | 716 |
| | Silt loam | 522 |
| | Silt | 0 |
| Fine to moderately fine-textured soil | | 1476 |
| | Clay loam | 514 |
| | Clay | 352 |
| | Silty clay loam | 253 |
| | Sandy clay loam | 226 |
| | Silty clay | 131 |
| | Sandy clay | 0 |




Table 7- Soil properties, number of data entries in the database (out of 5023 soil water infiltration curves in total),
and their statistical description

| Soil properties | Availability | Fr (%) | Mean | Min | Max | Median | CV (%) |
|---|---|---|---|---|---|---|---|
| Clay (%) | 3842 | 76 | 24 | 0 | 80 | 20 | 64 |
| Silt (%) | 3842 | 76 | 36 | 0 | 82 | 37 | 52 |
| Sand (%) | 3842 | 76 | 41 | 1 | 100 | 38 | 63 |
| Bulk density (g cm$^{-3}$) | 3295 | 66 | 1.32 | 0.14 | 2.81 | 1.35 | 20 |
| Organic carbon (%) | 3102 | 62 | 3 | 0 | 88 | 1 | 200 |
| Saturated hydraulic cond. (cm h$^{-1}$) | 1895 | 38 | 41 | 0 | 3004 | 3 | 426 |
| Initial soil water content (cm$^3$ cm$^{-3}$) | 1569 | 31 | 0.17 | 0 | 0.63 | 0.14 | 68 |
| Saturated soil water content (cm$^3$ cm$^{-3}$) | 1400 | 28 | 0.44 | 0.01 | 0.87 | 0.45 | 24 |
| Carbonate calcium equivalent (%) | 1399 | 28 | 14 | 0 | 56 | 8 | 101 |
| Electrical conductivity (dS m$^{-1}$) | 1113 | 22 | 25 | 0 | 358 | 1 | 249 |
| pH | 1081 | 22 | 7.4 | 4.7 | 8.6 | 7.6 | 12 |
| Particle density (g cm$^{-3}$) | 438 | 9 | 2.52 | 1.73 | 2.97 | 2.56 | 9 |
| Gypsum (%) | 380 | 8 | 4 | 0 | 49 | 3 | 137 |
| Cation exchange capacity (cmol$_c$ kg$^{-1}$) | 357 | 7 | 17 | 3 | 26 | 18 | 21 |
| Wet-aggregate stability (%) | 309 | 6 | 61 | 5 | 96 | 63 | 37 |
| Residual soil water content (cm$^3$ cm$^{-3}$) | 263 | 5 | 0.10 | 0.001 | 0.38 | 0.06 | 86 |
| Mean weight diameter (mm) | 258 | 5 | 1 | 0.10 | 2.75 | 1.0 | 54 |
| Gravel (%) | 243 | 5 | 18 | 0 | 92 | 15 | 84 |
| Sodium adsorption ratio | 156 | 3 | 5 | 0 | 89 | 1 | 351 |
| Soil water content at FC (cm$^3$ cm$^{-3}$) | 74 | 1 | 0.28 | 0.12 | 0.54 | 0.27 | 34 |
| Soil water content at PWP (cm$^3$ cm$^{-3}$) | 64 | 1 | 0.18 | 0.05 | 0.36 | 0.20 | 47 |
| Geometric mean diameter (mm) | 73 | 1 | 0.6 | 0.4 | 0.8 | 0.6 | 18 |

Fr: Frequency (%), Min: Minimum, Max: Maximum, CV: coefficient of variation.

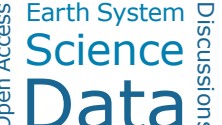


Table 8- Instruments used to measure soil infiltration curves

| Instrument/method used | | Infiltration curves |
|---|---|---|
| Ring | Double ring | 828 |
| | Single ring | 570 |
| | Beerkan (BEST) | 197 |
| Overall | | 1595 |
| Infiltrometer | Disc | 607 |
| | Mini-disc | 1140 |
| | Micro-disc | 36 |
| | Hood | 23 |
| | Tension | 752 |
| Overall | | 2558 |
| Permeameter | Guelph | 181 |
| | Aardvark | 50 |
| Overall | | 231 |
| Rainfall simulator | | 374 |
| Linear source method | | 10 |
| Point source method | | 4 |
| Not reported | | 251 |
| | Sum | 5023 |




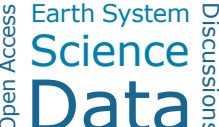


Table 9- Number of infiltration curves with a given land use types

| Land use | n | Land use | n |
|---|---|---|---|
| Agriculture | 2019 | Vineyards | 22 |
| Grassland | 821 | Upland | 11 |
| Pasture | 229 | Pure Sand | 10 |
| Forest | 204 | Brushwood | 6 |
| Garden | 152 | Road | 5 |
| Bare | 99 | Agro-pastoral | 4 |
| Urban Soils | 82 | Park | 3 |
| Savanna | 41 | Salt-marsh soil | 3 |
| Abandoned farms | 39 | Afforestation | 2 |
| Idle | 32 | Campus | 2 |
| Shrub | 30 | Residential | 2 |
| Available | 3818 | Unknown | 1205 |


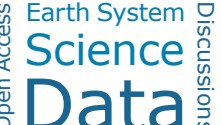

Table 10- Accuracy analysis of empirical models fitted to experimental data of infiltration

| Infiltration type | n | $R^2$ | | | | RMSE (cm) | | | | $R^2 >$ 0.90 | $R^2$ >0.99 |
|---|---|---|---|---|---|---|---|---|---|---|---|
| | | Mean | Min | Max | STD | Mean | Min | Max | STD | | |
| 1D | 828 | 0.985 | 0.529 | 1 | 0.049 | 0.900 | 1.3e-4 | 69.30 | 3.31 | 801 | 640 |
| 3D | 3350 | 0.975 | 0.032 | 1 | 0.066 | 0.449 | 5.5e-12 | 98.95 | 2.95 | 3136 | 2276 |
| All | 4178 | 0.977 | 0.032 | 1 | 0.063 | 0.538 | 5.5e-12 | 98.95 | 3.03 | 3937 | 2916 |

STD: standard deviation



Table 11- Estimated or measured average values of infiltration parameters for different textural classes extracted from the current database

| Texture class | Estimated by Eq. (8) or (9) | | | | | | | Measured | | | | Independent T test between measured and estimated $K_{sat}$ | |
| | $n^{§}$ | $S$ (cm h$^{-0.5}$) | | | $K_{sat}$ (cm h$^{-1}$) | | | $n^{§}$ | $K_{sat}$ (cm h$^{-1}$) | | | df | T value |
| | | Mean | Median | STD | Mean | Median | STD | | Mean | Median | STD | | |
| Sand | 291 | 2.3 | 0.26 | 4.3 | 42.2 | 15 | 134.5 | 229 | 43.6 | 24 | 149 | 518 | 0.10[ns] |
| Loamy sand | 92 | 10.6 | 5.7 | 17.5 | 61.4 | 10 | 173.2 | 63 | 24.6 | 8.2 | 72 | 153 | 1.59[ns] |
| Sandy loam | 500 | 9.2 | 2.95 | 15.7 | 32 | 3.1 | 94.5 | 424 | 41.2 | 5.7 | 166 | 922 | 1.05[ns] |
| Silt loam | 409 | 9.4 | 1.5 | 19.1 | 26.5 | 1.7 | 61.7 | 165 | 2.9 | 0.96 | 5.1 | 572 | 4.90[**] |
| Loam | 583 | 7.9 | 2.4 | 12.9 | 7.8 | 0.28 | 26.7 | 270 | 4.9 | 1.18 | 13.7 | 851 | 1.69[ns] |
| Sandy clay loam | 185 | 5.9 | 2.1 | 8.6 | 7.4 | 1.4 | 12.8 | 84 | 5.4 | 2.24 | 6.9 | 267 | 1.35[ns] |
| Silty clay loam | 250 | 3.2 | 0.64 | 12.5 | 10.6 | 1.7 | 24.1 | 64 | 12.3 | 2.42 | 63.2 | 312 | 0.32[ns] |
| Clay loam | 467 | 6.8 | 2.1 | 13.6 | 8.3 | 2.3 | 20 | 166 | 7.6 | 2.97 | 21.3 | 631 | 0.38[ns] |
| Sandy clay | - | - | - | - | - | - | - | - | - | - | - | - | - |
| Silty clay | 121 | 7.7 | 2.2 | 13.4 | 26.2 | 7.8 | 61.5 | 54 | 44.8 | 6.97 | 88.2 | 173 | 1.59[ns] |
| Clay | 333 | 14.6 | 1.7 | 39.5 | 354.3 | 1.3 | 1268.5 | 79 | 148.8 | 2.94 | 458.4 | 410 | 1.42[ns] |
| Silt | - | - | - | - | - | - | - | - | - | - | - | - | - |
| Total | 4179 | 8.5 | 2.6 | 18.2 | 46 | 1.8 | 374.8 | 1895 | 41 | 3.4 | 174 | - | - |

§: the number soils included in calculation
ns: insignificant and **: significant at 1 % probability level
STD: standard deviation

Table 12- Comparison of the estimated $K_{sat}$ values from current database (SWIG) with measured $K_{sat}$ values presented in literature

| Texture class | Data source | Clapp and Hornberger (1978) | Rosetta3 (Zhang and Schaap, 2017) | Cosby et al. (1984) | Rawls database (Schaap and Leij, 1998) | Ahuja database (Schaap and Leij, 1998) | UNSODA database (Schaap and Leij, 1998) | US soils $K_{sat}$ data (Pachepsky and Park, 2015) | EU-HYDI database (Weynants et al., 2013) |
|---|---|---|---|---|---|---|---|---|---|
| | | $K_{sat}$ (cm min$^{-1}$) | $logK_{sat}$/STD (cm day$^{-1}$) | $logK_{sat}$/STD (in h$^{-1}$) | $logK_{sat}$/STD (cm day$^{-1}$) | $logK_{sat}$/STD (cm day$^{-1}$) | $logK_{sat}$/STD (cm day$^{-1}$) | $logK_{sat}$/STD (cm h$^{-1}$) | $logK_{sat}$/STD (cm day$^{-1}$) |
| Sand | Literature | 1.056 | 2.81/0.59 (253) | 0.82/0.39 | 2.71/0.51 (97) | 3.01/0.45 (82) | 2.70/074 (129) | 1.57/0.71 (115) | 0.71/1.45 (264) |
| Sand | SWIG | 0.704 | 3.01 /3.51 (291) | 1.22 /1.73 | 3.01 /3.51 (291) | 3.01 /3.51 (291) | 3.01 /3.51 (291) | 1.63 /2.13 (291) | 3.01 /3.51 (291) |
| Loamy sand | Literature | 0.938 | 2.02/0.64 (167) | 0.30/0.51 | 1.91/0.61 (135) | 2.09/0.69 (19) | 2.36/0.59 (51) | 1.03/0.42 (76) | 0.80/1.41 (234) |
| Loamy sand | SWIG | 1.033 | 3.17 /3.63 (92) | 1.39 /1.84 | 3.17 /3.63 (92) | 3.17 /3.63 (92) | 3.17 /3.63 (92) | 1.79 /2.25 (92) | 3.17 /3.63 (92) |
| Sandy loam | Literature | 0.208 | 1.58/0.67 (315) | -0.13/0.67 | 1.53/0.65 (337) | 1.73/0.64 (65) | 1.58/0.92 (79) | 0.66/0.54 (169) | 1.17/1.34 (825) |
| Sandy loam | SWIG | 0.534 | 2.89 /3.36 (500) | 1.10 /1.58 | 2.89 /3.36 (500) | 2.89 /3.36 (500) | 2.89 /3.36 (500) | 1.51 /1.98 (500) | 2.89 /3.36 (500) |
| Silt loam | Literature | 0.043 | 1.28/0.74 (130) | -0.4/0.55 | 1.04/0.54 (217) | 1.24/0.47 (12) | 1.48/0.86 (103) | 0.11/0.87 (215) | 0.89/1.45 (714) |
| Silt loam | SWIG | 0.442 | 2.80 /3.17 (409) | 1.02 /1.39 | 2.80 /3.17 (409) | 2.80 /3.17 (409) | 2.80 /3.17 (409) | 1.42 /1.79 (409) | 2.80 /3.17 (409) |
| Loam | Literature | 0.042 | 1.09/0.92 (117) | -0.32/0.63 | 0.99/0.63 (137) | 0.83/0.95 (50) | 1.58/0.92 (62) | 0.12/0.79 (81) | 1.69/1.76 (411) |
| Loam | SWIG | 0.129 | 2.27 /2.81 (583) | 0.49 /1.02 | 2.27 /2.81 (583) | 2.27 /2.81 (583) | 2.27 /2.81 (583) | 0.89 /1.43 (583) | 2.27 /2.81 (583) |
| Sandy clay loam | Literature | 0.038 | 1.14/0.85 (13) | -0.2/0.54 | 1.29/0.71 (104) | 0.81/0.80 (36) | 0.99/1.21 (41) | 0.12/0.94 (139) | 0.73/1.45 (128) |
| Sandy clay loam | SWIG | 0.124 | 2.25 /2.49 (185) | 0.47 /0.70 | 2.25 /2.49 (185) | 2.25 /2.49 (185) | 2.25 /2.49 (185) | 0.87 /1.11 (185) | 2.25 /2.49 (185) |
| Silty clay loam | Literature | 0.010 | 1.04/0.74 (46) | -0.54/0.61 | 0.87/0.55 (47) | 1.09/0.78 (21) | 1.14/0.85 (21) | -0.15/0.75 (83) | 0.35/1.50 (364) |
| Silty clay loam | SWIG | 0.178 | 2.41 /2.77 (250) | 0.62 /0.98 | 2.41 /2.77 (250) | 2.41 /2.77 (250) | 2.41 /2.77 (250) | 1.03 /1.39 (250) | 2.41 /2.77 (250) |
| Clay loam | Literature | 0.015 | 0.87/1.11 (58) | -0.46/0.59 | 0.67/0.58 (77) | 0.79/1.08 (48) | 1.84/0.89 (25) | -0.03/0.94 (109) | 1.10/1.54 (284) |
| Clay loam | SWIG | 0.139 | 2.30 /2.68 (467) | 0.52 /0.90 | 2.30 /2.68 (467) | 2.30 /2.68 (467) | 2.30 /2.68 (467) | 0.92 /1.3 (467) | 2.30 /2.68 (467) |
| Sandy clay | Literature | 0.013 | 1.06/0.89 (10) | 0.01/0.33 | 1.33/0.33 (9) | -0.03/1.28 (2) | - (-) | -0.77/1.22 (21) | 0.81/1.56 (5) |
| Sandy clay | SWIG | - | - /- (-) | -/- | - /- (-) | - /- (-) | - /- (-) | - /- (-) | - /- (-) |
| Silty clay | Literature | 0.006 | 0.98/0.58 (14) | -0.72/0.69 | 0.82/0.55 (12) | 1.15/0.16 (5) | 0.92/0.71 (12) | -0.72/0.95 (22) | 0.18/1.32 (349) |
| Silty clay | SWIG | 0.439 | 2.80 /3.17 (121) | 1.02 /1.39 | 2.80 /3.17 (121) | 2.80 /3.17 (121) | 2.80 /3.17 (121) | 1.42 /1.79 (121) | 2.80 /3.17 (121) |
| Clay | Literature | 0.008 | 1.17/0.92 (60) | - | 0.94/0.31 (34) | 1.03/0.83 (31) | 1.41/015 (27) | -0.17/0.71 (115) | -0.08/1.41 (737) |
| Clay | SWIG | 5.906 | 3.93 /4.48 (333) | 2.15 /2.70 | 3.93 /4.48 (333) | 3.93 /4.48 (333) | 3.93 /4.48 (333) | 2.55 /3.10 (333) | 3.93 /4.48 (333) |
| Silt | Literature | - | 1.64/0.27 (3) | - | 1.43/- (3) | - (-) | 1.75/0.20 (3) | - (-) | -0.29/1.56 (11) |
| Silt | SWIG | - | -/- (-) | -/- | -/- (-) | -/- (-) | -/- (-) | -/- (-) | -/- (-) |




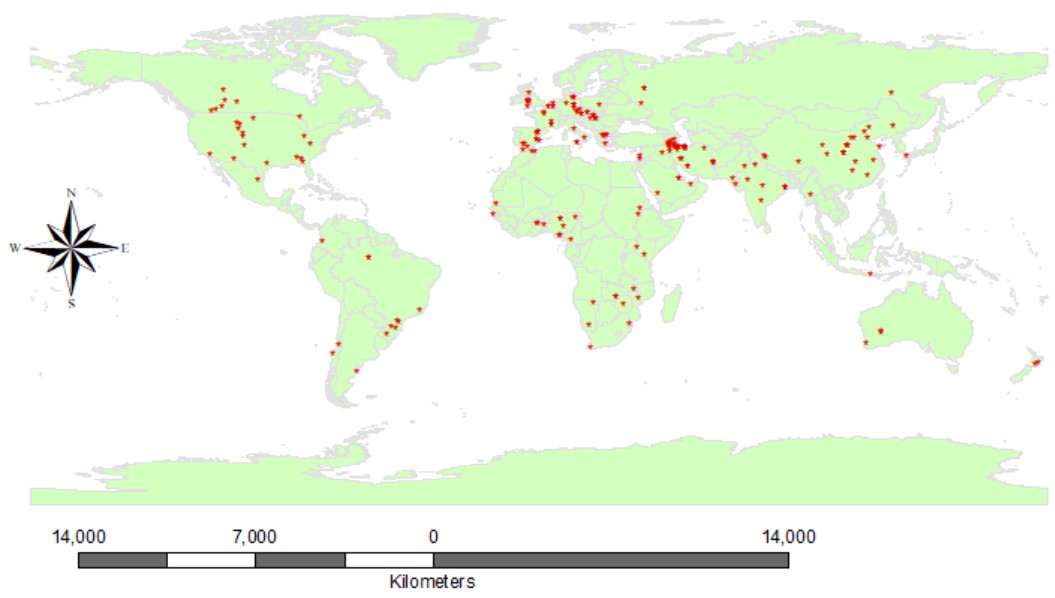


Figure 1- Global distribution of infiltration measuring sites that were included in the database





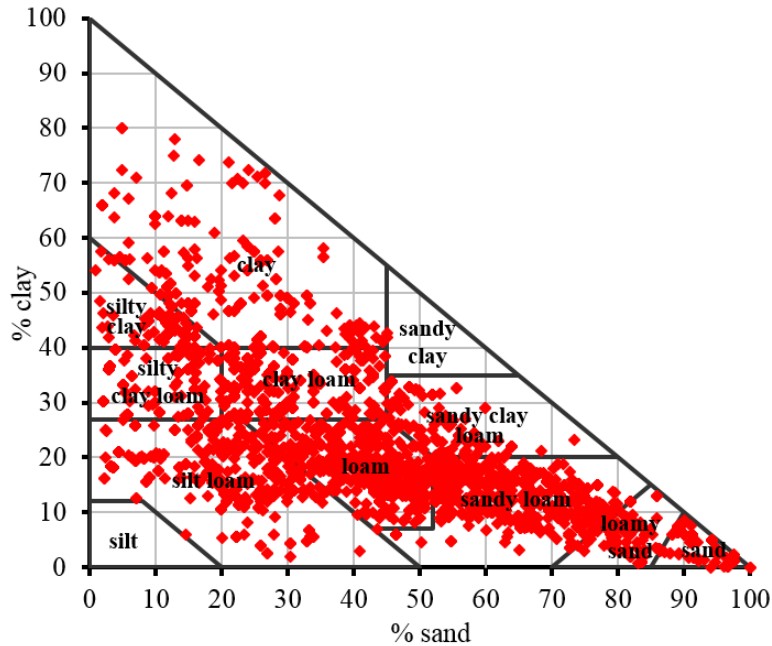


Figure 2 - Textural distribution of soils (plotted on USDA textural triangle) for which infiltration data are included

in the database.



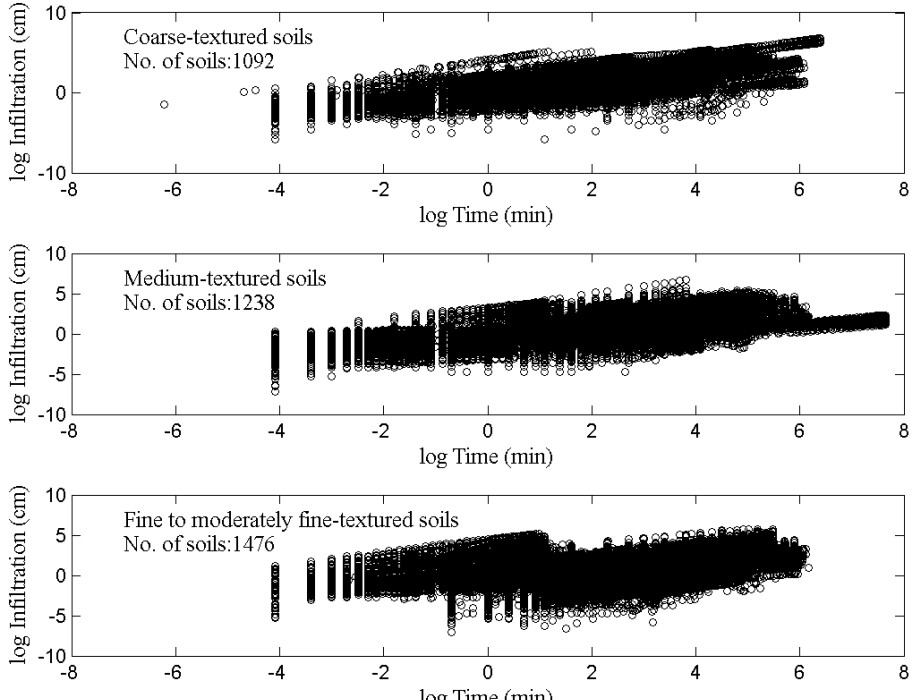


Figure 3- Cumulative infiltration curves for the three identified textural groups: coarse (sand, loamy sand, and sandy

loam), medium (loam, silt loam, silt), and fine to moderately fine (sandy clay, sandy clay loam, clay loam, sandy

clay loam, silty clay, clay)



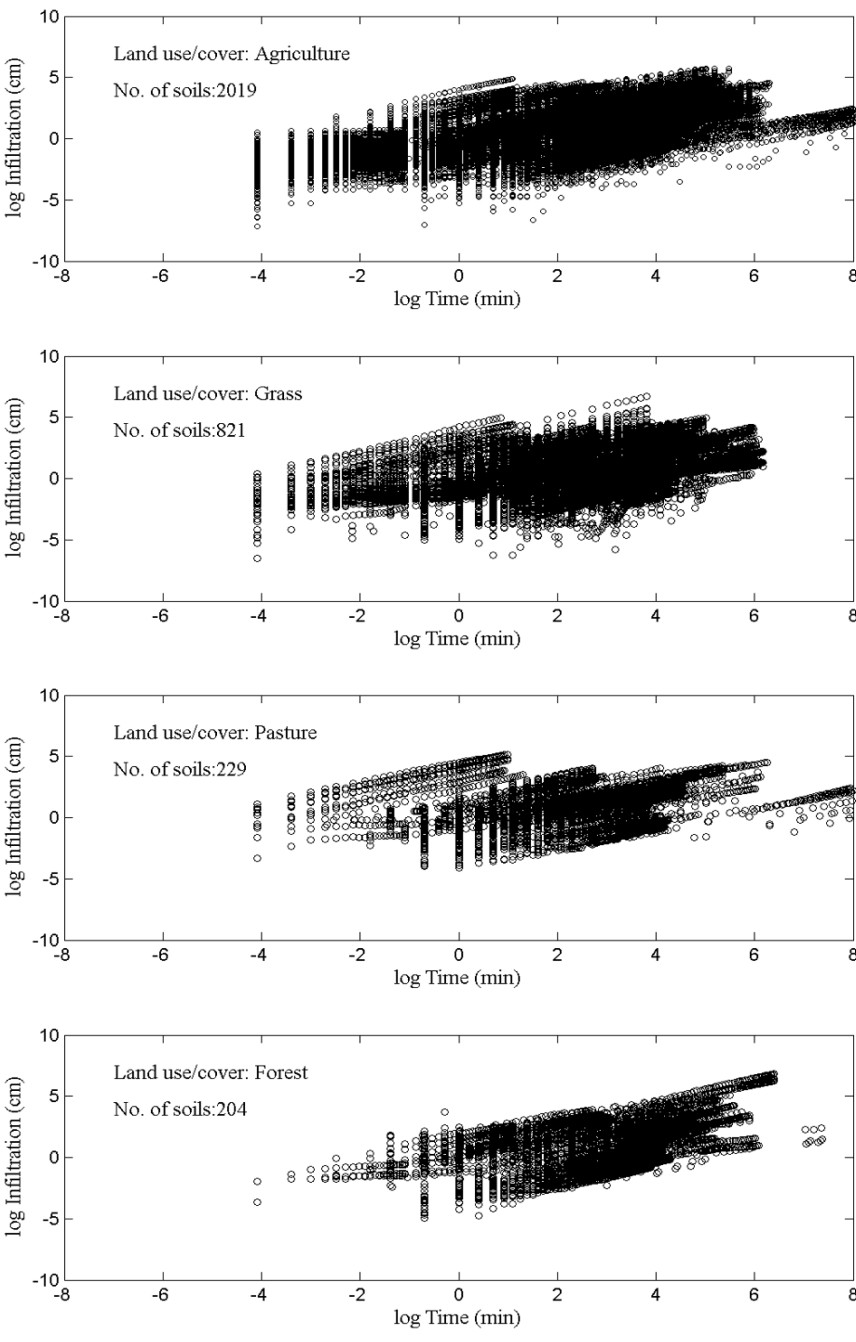


Figure 4- Cumulative infiltration curves for the four dominant land use types in examined sites





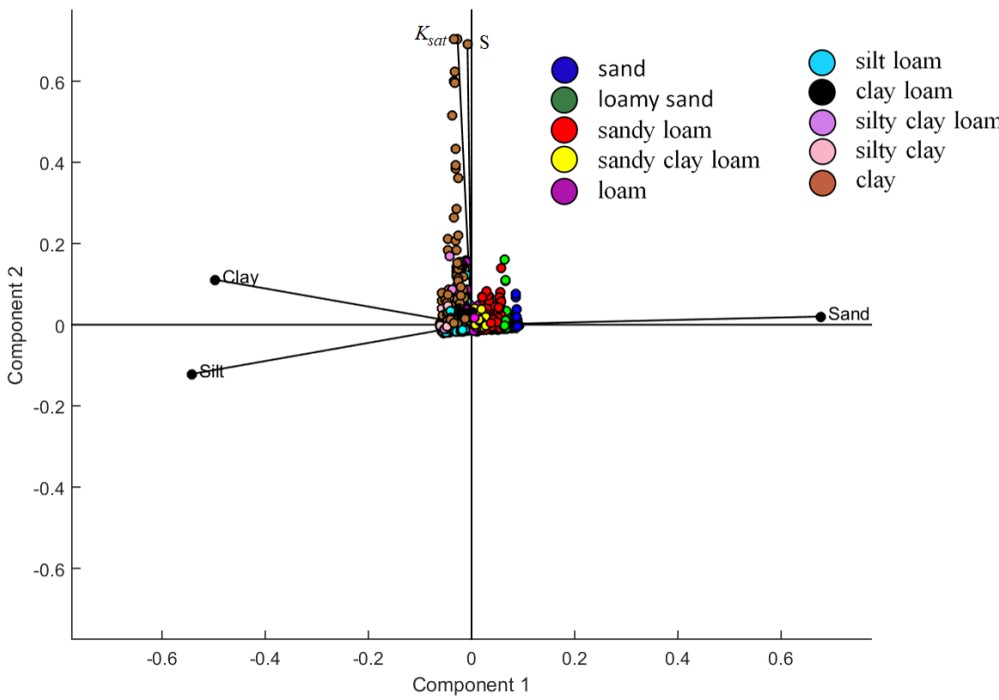


Figure 5- The relationships between clay, silt, sand contents and estimated hydraulic parameters (S and Ksat)





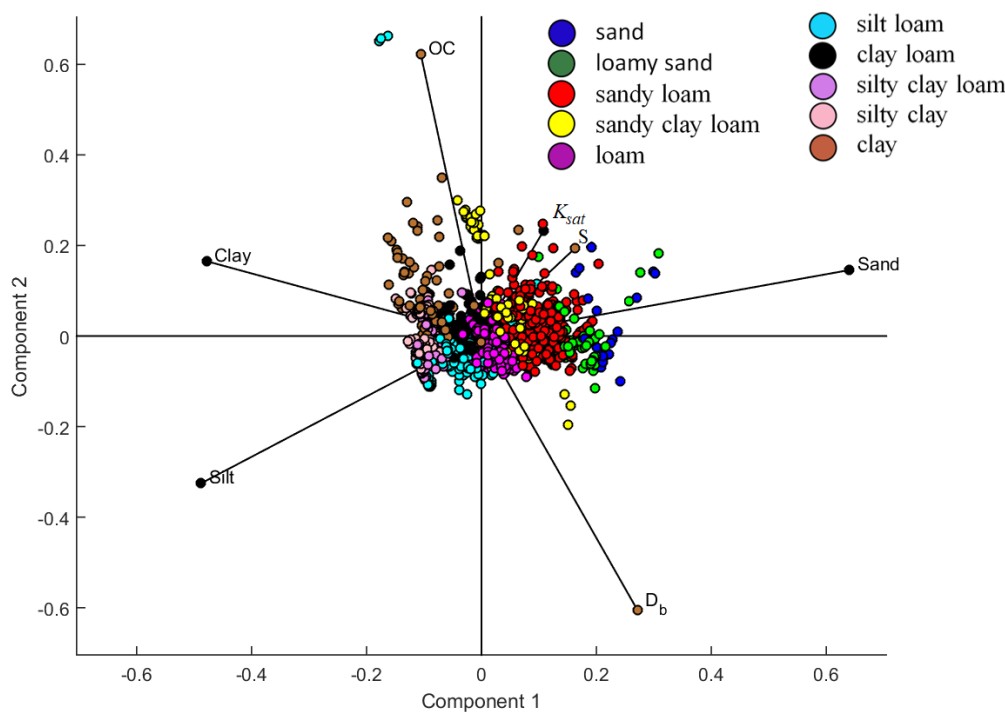


Figure 6- The relationships between clay, silt, sand contents, Db, and OC and estimated hydraulic parameters (S and

$K_{sat}$)
