# Peer review of "Development and Analysis of Soil Water Infiltration Global Database"

_Earth System Science Data, 2018_

## Short Comment (SC1) · 25 Mar 2018

Through a global survey of availability of soil infiltration data, the main authors of this manuscript developed a general accessible global data base of some 5000+ infiltration data sets, with all those who submitted data recognized through co-authorship. The database includes additional supplemental data such as soil texture, SOM, bulk density, saturated hydraulic conductivity, EC and pH, and landuse, if available. All data were digitized, and subsequently fitted through analytical 1D or 3D analytical solutions, providing for fitted values of sorptivity (S) and saturated conductivity (Ksat). Subsequently, the paper analyzed and compared measured with fitted Ksat values, and used principal component analysis (PCA) to analyze and discuss possible relationships between soil properties with S and/or Ksat. However, as also stated by the authors, the

performed analyses are limited and much more could have been donem , but that was not their objective. Moreover, the authors emphasize the many uncertainties associated with the various measurements and model assumptions. Despite that, the manuscript highlights the potential uses of this database for future research, as well as the need to expand the database, as various world regions are under-represented.

In all, this reviewer was impressed with the commitment of the main authors to provide such a accessible global soil infiltration database. My main other comments are: 1. For those many readers that are likely not well-versed in soil infiltration, its limitations in measurement and modeling, it would be best if a literature reference was provided. I could give an example of that, but ask the authors to contact me, if they are interested. 2. Indeed, the main discussion is on data uncertainty, for example on the discrepancies between independently measured and fitted Ksat values, and the lack of correlation with soil texture. The authors provide various reasons, including the scale of measurement and differences between field-measured and lab-measured infiltration data. However, I would pose that much of that is related to the lack of including soil structural information, such as macro porosity quantification or other possible soil attributes. I think that this manuscript deserves such discussion, so readers are aware. Moreover, it could aspire others to collect such information when conducting additional soil infiltration measurements, and can include this in the database in the future.

---

## Short Comment (SC2) · 16 Apr 2018

The authors present a 5000+ sized international data collection of soil infiltration measurements and related meta-information. I congratulate the authors for pooling these data into a great database and providing an initial exploration of the data collection. It is exemplary that such amount of data were openly contributed and made available to the community with no limitations. In my view, the lack of pooled field-based soil hydraulic data constituted a large knowledge gap for a long time now. I have a few questions about methodology and a number of small comments on the text, but I think this database and its documentation yield great service to the international soil community. I look forward to seeing the final version and further analyses performed on this data set.

[Figure]

I have three questions about methodology or its documentation.

Particle-size distribution (PSD): It is not surprising that the authors did not find much correlation between PSD and Ksat/S. Yet, I find it important to be clear about the way soil texture data were handled. Internationally, there are typically more than one PSD standards followed (e.g. USDA/FAO vs. IUSS), yet there appears to be only USDA/FAO conform data reported. Yet other systems may not even be possible to convert/interpolate, due to working with a fraction other than <2mm. Please add necessary information on how non-USDA/FAO-conform PSD data were handled. Were those rejected? Interpolated in any ways? Is raw data available? (L269-284, and Table 4)

Field capacity (FC) although available only for a limited number of cases: This is another example where international standards do not match – there are at least 5 matric potentials at which FC is approximated. Is there information on what definition was reported, and was there any opportunity to standardize – or at least provide metadata - if reported at different matric potentials?

Other properties, for example BD, Ksat or saturated water content: Is there any information on their methodology? Core method vs. clod method? Field or lab Ksat (constant vs. falling head?) or fitted?, sensory or gravimetry? If such information is not available, I recommend that it is stated that those were not collected or provided. I expect that methodology on Ksat will especially be of interest.

I think the above should at least be commented on in the paper – or described where possible – to help avoid misinterpretations or the lack of information may hinder the database's use in any other ways.

Minor comments, editorials:

L215: made on samples of. . .

L290: Sparse coverage?
L301: Since this is often the same for other large data collections, I suggest replacing the end o fthe sentence as: "...(Fig 2), which makes SWIG a valuable data source for comprehensive studies."

L302-303: Rephrase to make it an independent sentence. (Skip 'because' and perhaps add that it would still be desirable to know about those soils.

L332: With 22%, grasslands are the second most frequently represented land use type.

L336: replace 'striking' with 'noticeable' or something similar. Alternatively: "Data show that the upper and lower...."

L349-350: Does this lumping originate from the cited paper? If not, please explain.

L356: Please spell out what is meant by 'this'

L358: replace 'rejected' with 'excluded'. ...were excluded from the presented study. The same in L359.

L363: the lsqnonlin

L365-366: restructure sentence: ...R2 values higher than 0.9 and 0.99 were obtained in 94 and 68% of the cases respectively."

L368: from the analysis

L379: from the SWIG

L387: replace 'striking' with a more objective sounding term. It was observed....or something similar.

L388-392: It would be useful to add a sentence or two here, summarizing what exactly constitute the problem. (e.g. sample size vs. representative elementary volume, sample excluding cracks or biopores, imperfection of sampling, etc.)

L397: was performed

L412: that the examined basic soil properties. . .

L416: replace 'done' with 'implemented'

L417: does not provide adequate means to estimate Ks

L421: databases

L425: errors

L427: difficult, since the required

L27-428: The uncertainty and variability related to the applied measurement technique . . .. may be assessed as information on the applied techniques is available.

L430: a strong effort has been made. . .

L431: . . .any probable error of this nature.

L435-438: Merge this para with the previous under the same umbrella with soil hydraulic properties. It is a very similar thought.

L440-441: Do you refer to measurement scale here? How about assumptions about initial conditions, 1D vs 3D flow, etc. . .? Acknowledge those other potential sources of uncertainty.

L443: please provide reference(s)

L444-454: First, the quality of text in this section should be improved in general. Second, I think it would be better to present these cases more in a general context, perhaps even with 2-3 references.

L475-476: . . .climate models, texture is not the main controlling factor.

L476: the SWIG database

References: Das Gupta should be Dasgupta. Please correct it in the relevant tables as well.

Figure 1: I don't think the scale is necessary for a World map, especially since part of the map is distorted.

Figures 3 and 4: There is a concern of visibility in these two figures. Also, there is very little difference that the reader can comprehend between the respective panels. They are not discussed too much either. I suggest that these two figures are removed, or some alternate way of presenting the relevant data is found.

Figures 5-6: What do the multiple points with the same color represent within a texture class? They appear too few (especially in Figure 5) to be individual samples.

―――――――――――――――

---

## Short Comment (SC3) · 18 Apr 2018

I as a coauthor of the manuscript admire reviewer's very valuable comments particularly on structural information necessary for better modeling of the infiltration processes. However, I can claim that the information and datasets presented here were the maximum available and accessible data which could have been provided by contributors from different parts of the world. Though, much more datasets are available throughout the globe which hopefully will be extended in future researches in similar fields.

---

## Author Comment (AC1) · 30 Apr 2018

Authors' Response to the Review Comments Journal: Earth Syst. Sci. Data Discuss Manuscript #: Earth Syst. Sci. Data Discuss., https://doi.org/10.5194/essd-2018-11 Title of Paper: Development and Analysis of Soil Water Infiltration Global Database Authors: Mehdi Rahmati et al.

We thank the reviewers and editor for their valuable comments to improve this manuscript. As far as possible the comments have been addressed in the revised version of the manuscript. Revised version of Manuscript (tracked and cleaned) + supplement are attached. Following are the replies for specific comments.

Comments from Prof. Jan Hopmans Comment: "Through a global survey of availability

of soil infiltration data, the main authors of this manuscript developed a general accessible global data base of some 5000+ infiltration data sets, with all those who submitted data recognized through co-authorship. The database includes additional supplemental data such as soil texture, SOM, bulk density, saturated hydraulic conductivity, EC and pH, and landuse, if available. All data were digitized, and subsequently fitted through analytical 1D or 3D analytical solutions, providing for fitted values of sorptivity (S) and saturated conductivity (Ksat). Subsequently, the paper analyzed and compared measured with fitted Ksat values, and used principal component analysis (PCA) to analyze and discuss possible relationships between soil properties with S and/or Ksat. However, as also stated by the authors, the performed analyses are limited and much more could have been done, but that was not their objective. Moreover, the authors emphasize the many uncertainties associated with the various measurements and model assumptions. Despite that, the manuscript highlights the potential uses of this database for future research, as well as the need to expand the database, as various world regions are under-represented. In all, this reviewer was impressed with the commitment of the main authors to provide such an accessible global soil infiltration database."

Response: Authors thank Prof. Dr. Jan Hopmans for giving such a positive feedback on this huge effort that has been made to collect these data.

Comment: "My main other comments are: 1. for those many readers that are likely not well-versed in soil infiltration, its limitations in measurement and modeling, it would be best if a literature reference was provided. I could give an example of that, but ask the authors to contact me, if they are interested."

Response: Actually, different methodologies to measure soil infiltration and their formulation and limitations are already provided in a supplement file. We had missed to refer the readers to this file. Couple of lines is added at the end of Introduction section to cover this. Several references are also provided.

Comment: 2. Indeed, the main discussion is on data uncertainty, for example on the discrepancies between independently measured and fitted Ksat values, and the lack of correlation with soil texture. The authors provide various reasons, including the scale of measurement and differences between field-measured and lab-measured infiltration data. However, I would pose that much of that is related to the lack of including soil structural information, such as macro porosity quantification or other possible soil attributes. I think that this manuscript deserves such discussion, so readers are aware. Moreover, it could aspire others to collect such information when conducting additional soil infiltration measurements, and can include this in the database in the future."

Response: This comment is discussing about the structure effect on soil infiltration that is addressed in manuscript. Please see lines 399-407.

Comments from Prof. Attila Nemes Comment: "The authors present a 5000+ sized international data collection of soil infiltration measurements and related meta-information. I congratulate the authors for pooling these data into a great database and providing an initial exploration of the data collection. It is exemplary that such amount of data was openly contributed and made available to the community with no limitations. In my view, the lack of pooled field-based soil hydraulic data constituted a large knowledge gap for a long time now. I have a few questions about methodology and a number of small comments on the text, but I think this database and its documentation yield great service to the international soil community. I look forward to seeing the final version and further analyses performed on this data set."

Response: Authors are grateful to Prof. Dr. Attila Nemes for his positive feedback. Such a positive feedback from him as a pioneer in supplying this kind of huge databases is unique and outstanding. We found the comments very useful.

Comment: "I have three questions about methodology or its documentation. Particle-size distribution (PSD): It is not surprising that the authors did not find much correlation between PSD and Ksat/S. Yet, I find it important to be clear about the way soil texture

data were handled. Internationally, there are typically more than one PSD standards followed (e.g. USDA/FAO vs. IUSS), yet there appears to be only USDA/FAO conform data reported. Yet other systems may not even be possible to convert/interpolate, due to working with a fraction other than <2mm. Please add necessary information on how non-USDA/FAO-conform PSD data were handled. Were those rejected? Interpolated in any ways? Is raw data available? (L269-284, and Table 4).

Field capacity (FC) although available only for a limited number of cases: This is another example where international standards do not match – there are at least 5 matric potentials at which FC is approximated. Is there information on what definition was reported, and was there any opportunity to standardize – or at least provide metadata - if reported at different matric potentials? Other properties, for example BD, Ksat or saturated water content: Is there any information on their methodology? Core method vs. clod method? Field or lab Ksat (constant vs. falling head?) or fitted? Sensory or gravimetric? If such information is not available, I recommend that it is stated that those were not collected or provided. I expect that methodology on Ksat will especially be of interest.

I think the above should at least be commented on in the paper – or described where possible – to help avoid misinterpretations or the lack of information may hinder the database's use in any other ways."

Response: The reviewer is right by this comment/question. This is what that some of our co-authors also had warned us about the mixture of different standards in measuring or calculating soil properties. Actually, no conversion has been made and only raw data are reported in database. We simply assumed that all soil properties, more specifically soil texture, are measured by routine USDA methodology. However, we have supplied the reference for all data (if available) to enable people to retrieve more information if needed. This is discussed through the manuscript to warn the readers being aware of this issue. Please refer to lines 457-465.

Comment: Minor comments, editorials: L215: made on samples of. . . L290: Sparse coverage? L301: Since this is often the same for other large data collections, I suggest replacing the end of the sentence as: (Fig 2), which makes SWIG a valuable data source for comprehensive studies." L302-303: Rephrase to make it an independent sentence. (Skip 'because' and perhaps add that it would still be desirable to know about those soils. L332: With 22%, grasslands are the second most frequently represented land use type. L336: replace 'striking' with 'noticeable' or something similar. Alternatively: "Data show that the upper and lower. . .." L349-350: Does this lumping originate from the cited paper? If not, please explain. L356: Please spell out what is meant by 'this' L358: replace 'rejected' with 'excluded'. . .. were excluded from the presented study. The same in L359. L363: the lsqnonlin L365-366: restructure sentence: . . .$R^2$ values higher than 0.9 and 0.99 were obtained in 94 and 68% of the cases respectively." L368: from the analysis L379: from the SWIG L387: replace 'striking' with a more objective sounding term. It was observed. . .. or something similar. L388-392: It would be useful to add a sentence or two here, summarizing what exactly constitute the problem. (e.g. sample size vs. representative elementary volume, sample excluding cracks or biopores, imperfection of sampling, etc.) L397: was performed L412: that the examined basic soil properties. . . L416: replace 'done' with 'implemented' L417: does not provide adequate means to estimate Ks L421: databases L425: errors L427: difficult, since the required L27-428: The uncertainty and variability related to the applied measurement technique . . .. may be assessed as information on the applied techniques is available. L430: a strong effort has been made. . . L431: . . .any probable error of this nature. L435-438: Merge this para with the previous under the same umbrella with soil hydraulic properties. It is a very similar thought. L440-441: Do you refer to measurement scale here? How about assumptions about initial conditions, 1D vs 3D flow, etc.. .? Acknowledge those other potential sources of uncertainty. L443: please provide reference(s) L444-454: First, the quality of text in this section should be improved in general. Second, I think it would be better to present these cases more in a general context, perhaps even with 2-3 references. L475-476: . . .climate models,

texture is not the main controlling factor. L476: the SWIG database References: Das Gupta should be Dasgupta. Please correct it in the relevant tables as well Figure 1: I don't think the scale is necessary for a World map, especially since part of the map is distorted.

Response: done

Comment: Figures 3 and 4: There is a concern of visibility in these two figures. Also, there is very little difference that the reader can comprehend between the respective panels. They are not discussed too much either. I suggest that these two figures are removed, or some alternate way of presenting the relevant data is found. Figures 5-6: What do the multiple points with the same color represent within a texture class? They appear too few (especially in Figure 5) to be individual samples.

Response: Figure 3 and 4 are removed now. Nothing more except the soil texture they show. By this coloring figure we were trying to illustrate that which texture class is relevant to which component and which soil properties.

Please also note the supplement to this comment:
https://www.earth-syst-sci-data-discuss.net/essd-2018-11/essd-2018-11-AC1-supplement.zip

---

## Referee Comment (RC2) · M Vanclooster (Referee) · 7 May 2018

Review ESSD-2018-11 : Development and Analysis of Soil Water Infiltration Global Database (Rahmati et al.)

M. Vanclooster

The paper describes the construction of the Soil Water Infiltration Global Database (SWIG). Based on a detailed literature research, as well the provision of data of the many co-authors of the manuscript, an infiltration experiment database is constructed that tends towards a global coverage. In total 5023 infiltration curves are compiled. In addition to the basic information related to the infiltration experiments, additional ancillary information is provided allowing to explain the infiltration. This allows constructing

explanatory and predictive infiltration models. Part of the infiltration experiments is analyzed using physical based infiltration models (1D and 3 D analytical infiltration equations). Finally, the authors discuss the strengths and the weaknesses of the database and the possible applications of the data in hydrological and environmental studies.

The presented work is novel, as no similar database of experimental infiltration curves exists with the ambition of a global coverage. This database may, therefore, play a key role in improving the parametrization of the infiltration process in global Earth System Models (ESM). This improvement is of major importance, as infiltration is a basic process controlling hydrological fluxes in earth systems but yet poorly represented. The database has, therefore, the potential to improve the parameterization of soil hydrological fluxes and to reduce the uncertainties associated with current soil hydraulic pedotransfer functions. The paper has, therefore, the potential to become a valuable contribution to ESSD. However, some concerns can be formulated that should be considered in a revision before the paper can be accepted as a full publication in ESSD. The major concerns can be formulated as follows:

1. The paper deals with infiltration, but a clear definition of infiltration is lacking. The authors generally refer in their manuscript to the infiltration that will be observed in controlled field experiments, without being explicit on this. Yet, infiltration is a more general hydrological process that occurs also in transient natural and uncontrolled conditions. Hence many statements referring to the controlled infiltration experiment will not hold for the uncontrolled natural infiltration process. This adds confusion in many statements in the paper (e.g. infiltration generally decreases in time...), that should be corrected by clearly defining the type of infiltration that is considered in the analysis.

2. The authors should more correctly define the extension scale of the database. The SWIG has the ambition to be global, but yet data were compiled from "only" 54 countries all over the world. It is not guaranteed that global soil variability is represented when collecting data from "only" 54 countries. The fact that all textural classes are nearly represented in the database does not warrant representativeness. It would be

better to evaluate other soil properties (e.g. major soil type according to FAO or WRB soil reference system), to demonstrate and claim global representativeness. The map in Figure 1 clearly shows that major regions of the world are not represented, which may considerably limit the global scope of the database or the application of data from the database in global Earth System Models. It may be suggested that the authors perform a representativity analysis, in which not only "countries" or "available texture class" are considered as a criterion for representativeness, but other criteria such as "WRB or FAO soil type", "earth climate region", "earth ecozone region".

3. A set of ancillary variables are introduced in the data set. The intention of this is to apply data mining techniques to explain the infiltration process parameters and hence to allow developing new explanatory or predictive models. However, the quality of these models will depend on the quality of the ancillary variables that have been introduced in the database. Unfortunately, some ancillary variables are proposed in Table 4 that are not well defined or not well normalized or standardized. The added value of adding these ancillary variables to the database should be reconsidered. This is particularly the case for FC (many definitions of the field water capacity can be retrieved in the literature, see e.g. Nachabe, 1998), soil pH (measured in water or in KCl), and wet-aggregate stability.

4. A very limited and preliminary data exploration analysis by means of PCA is presented in section 3.6. The preliminary and limited scope of this analysis questions the overall results of this analysis. For instance, the sorptivity, S, has been integrated into the PCA analysis. Yet S is not an intrinsic time-invariant soil property, but a soil variable that strongly is affected by the initial and saturated water content. Mixing such time dynamic state variable with static properties (such as soil texture, Ks, ...) in a PCA has little sense, as the results will strongly depend on the initial water content before the infiltration will start.

In addition to this major concerns, some minor concerns are listed below

Line 170. "In addition to its global coverage". Cf. above. Global coverage should be demonstrated by a representativity analysis.

Line 176. We should expect that land use can be assessed for all the cases. If the spatial coordinates of the infiltration data are known, land use can be retrieved from historical land use data archives (see Google Earth Engine).

Line 199. "In general, the soil infiltration rate decreases nonlinearly over time". Cf above. This is specifically the case when the infiltration process is studied under controlled conditions (typically as the cases where controlled infiltration experiments are performed). In general, infiltration is very time dynamic, conditioned to time variable climatic conditions, and in-situ infiltration rates will not 'in general' decrease with time. It is therefore suggested to give a clear definition of "infiltration" in this paper and make a clear distinction between in-situ and controlled experimental infiltration processes.

Line 207. "However, as infiltration proceeds, the gradient….." This is only the case when a pressure head boundary condition is used to define the infiltration process. This may not be the case when flux boundary conditions are used (e.g. constant precipitation). For instance, in case infiltration in a dry soil is analyzed subjected to constant flux boundary conditions, with an imposed flux that is smaller than the saturated hydraulic conductivity of the soil, than no ponding will occur, all water will infiltrate, and no decrease of pressure gradients will be observed.

Line 209. "…approximates saturated hydraulic conductivity". This definition is often debated in the literature. For instance, Kutilek and Nielsen (1994), suggest Ks = 2/3 * the asymptotic value.

Line 218. The Richards equation written in water content form is often referred to as the Fokker-Planck water diffusion equation.

Line 245. ".. from all over the globe". Be more rigorous. Many parts of the globe have not been considered for data collection.

[Figure]

Line 278. To avoid confusion, define Mi exactly.

Line 302. Please reformulate this phrase (what is the principle phrase?).

Line 327. This statement is clearly not supported by the data in Figure 4. Please avoid general statements that are not supported by the data, or introduce cautionary notes to put such statements in a correct perspective.

Line 363. Correct: "Matlab$^{TM}$".

Line 372. The "material and methods" section does not explain in detail the difference between those two approaches. What is exactly meant by 'measured Ksat'?

Line 888. If Table 3 is the continuation of Table 2, then it should not be a new table. (So no increase in table number).

Line 911. For improving the comparability, please harmonized the data in the same units (eg. log values in cm/day).

Line 918. Figure 3 adds very little information to the manuscript and is not very useful for the reader. Please consider to eliminate.

References: Kutílek, Miroslav, and Donald R. Nielsen. Soil hydrology: textbook for students of soil science, agriculture, forestry, geoecology, hydrology, geomorphology and other related disciplines. Catena Verlag, 1994.

Nachabe, M. H. "Refining the definition of field capacity in the literature." Journal of irrigation and drainage engineering 124.4 (1998): 230-232.

---

## Author Comment (AC2) · 19 May 2018

Authors' Response to the Review Comments Journal: Earth Syst. Sci. Data Discuss Manuscript #: Earth Syst. Sci. Data Discuss., https://doi.org/10.5194/essd-2018-11 Title of Paper: Development and Analysis of Soil Water Infiltration Global Database Authors: Mehdi Rahmati et al.

We thank the reviewer (Prof. Dr. Marnik Vanclooster) for his valuable comments to improve this manuscript. As far as possible the comments have been addressed in the revised version of the manuscript. Following are the replies for specific comments.

Comment: The paper describes the construction of the Soil Water Infiltration Global Database (SWIG). Based on a detailed literature research, as well the provision of

data of the many co-authors of the manuscript, an infiltration experiment database is constructed that tends towards a global coverage. In total 5023 infiltration curves are compiled. In addition to the basic information related to the infiltration experiments, additional ancillary information is provided allowing to explain the infiltration. This allows constructing explanatory and predictive infiltration models. Part of the infiltration experiments is analyzed using physical based infiltration models (1D and 3 D analytical infiltration equations). Finally, the authors discuss the strengths and the weaknesses of the database and the possible applications of the data in hydrological and environmental studies. The presented work is novel, as no similar database of experimental infiltration curves exists with the ambition of a global coverage. This database may, therefore, play a key role in improving the parametrization of the infiltration process in global Earth System Models (ESM). This improvement is of major importance, as infiltration is a basic process controlling hydrological fluxes in earth systems but yet poorly represented. The database has, therefore, the potential to improve the parameterization of soil hydrological fluxes and to reduce the uncertainties associated with current soil hydraulic pedotransfer functions. The paper has, therefore, the potential to become a valuable contribution to ESSD. However, some concerns can be formulated that should be considered in a revision before the paper can be accepted as a full publication in ESSD. The major concerns can be formulated as follows:

Response: Authors thank Prof. Dr. Marnik Vanclooster for his positive feedback on this work. We understand the reviewers concerns about the presentation of the collected data and then we did our best to address nearly all of his concerns in manuscript.

Comment: 1. The paper deals with infiltration, but a clear definition of infiltration is lacking. The authors generally refer in their manuscript to the infiltration that will be observed in controlled field experiments, without being explicit on this. Yet, infiltration is a more general hydrological process that occurs also in transient natural and uncontrolled conditions. Hence many statements referring to the controlled infiltration experiment will not hold for the uncontrolled natural infiltration process. This adds con-
fusion in many statements in the paper (e.g. infiltration generally decreases in time: :), that should be corrected by clearly defining the type of infiltration that is considered in the analysis. Response: The authors thank the reviewer for this accurate comment. The revised paper was modified accordingly to implement more details about water infiltration at the scale of the water cycle and water infiltration as monitored on the field. Please refer to lines 199-216 and 223-227.

Lines 199-216: Two main mechanisms are responsible for the generation of excess water that produce overland flow: Dunne saturation excess and Hortonian infiltration excess (Sahoo et al., 2008). Dunne overland flow or saturation excess occurs when the soil profile is completely saturated and precipitation can no longer infiltrate into soil. The Dunne mechanism is more common to near-channel areas or it is generated from partial areas of the hillslope where water tables are shallowest (Sahoo et al., 2008). On the other hand, Hortonian overland flow is characterized by rainfall intensities exceeding the infiltration rate of the soil. In the other words, during a rainfall event, water infiltration at the soil surface and runoff are highly conditioned by the boundary condition, namely, under field condition, of the rainfall intensity and the soil hydraulic properties. If the rainfall intensity is lower than the soil infiltrability, water will completely infiltrate into the soil without any runoff (Hillel, 2013). In this case, the infiltration rate will align with the rainfall intensity. Otherwise, if the precipitation intensity exceeds the soil infiltration rate at a certain moment in time, excess water will be generated even if the soil profile is unsaturated. In this case water will pond on the soil surface and becomes available for surface runoff. In this case, the boundary condition at soil surface shifts from imposed flow rate to imposed water pressure head. Admitting that the water pressure heads remain constant at the soil surface, the infiltration rate describes a decreasing function over time and tending towards the value of the hydraulic conductivity corresponding to the water pressure head at the surface (Angulo et al., 2016, Chow et al., 1988). In the past decades, water infiltration tests, using either ponded or tension infiltrometers have been developed to quantify the cumulative infiltration at the soil surface. In this case, the 3D axisymmetric water infiltration corresponds to an upper boundary defined by a constant water pressure head or a series of constant water pressure heads.

Lines 223-227: As stated above, the infiltration rate i(t) is expected to strongly decrease down to a plateau defined by the value of the hydraulic conductivity corresponding to the imposed water pressure head plus a term related to radial water infiltration (Angulo et al., 2016). In the case of large rings, the final infiltration rate approaches the value of the hydraulic conductivity corresponding to the imposed water pressure head (gravity flow). Consequently, if water ponding is imposed at surface, i(t) tends towards the saturated hydraulic conductivity.

Comment: 2. The authors should more correctly define the extension scale of the database. The SWIG has the ambition to be global, but yet data were compiled from "only" 54 countries all over the world. It is not guaranteed that global soil variability is represented when collecting data from "only" 54 countries. The fact that all textural classes are nearly represented in the database does not warrant representativeness. It would be better to evaluate other soil properties (e.g. major soil type according to FAO or WRB soil reference system), to demonstrate and claim global representativeness. The map in Figure 1 clearly shows that major regions of the world are not represented, which may considerably limit the global scope of the database or the application of data from the database in global Earth System Models. It may be suggested that the authors perform a representatively analysis, in which not only "countries" or "available texture class" are considered as a criterion for representativeness, but other criteria such as "WRB or FAO soil type", "earth climate region", "earth ecozone region".

Response: The data that are provided is the best that can be done at present to make available infiltration data with the largest possible coverage. As these data are spread over all continents it is a fair to state that SWIG aims at providing global coverage of infiltration data. Of course it is possible to remap the coordinates of the infiltration experiments onto other spatial attributes eventually pointing out gaps in covering soil orders or climatic zones. But this does not question the usefulness of the database

provided. We are also not claiming that we can cover global soil variability and thus provide a full picture of global variability in soil infiltration properties. As said, this is the best that can be done at present and we are aiming/hoping to collect more data in future to release the second version of SWIG that will contain more data from remaining countries. We also analyzed the number of infiltration curves available for the different Köppen−Geiger Climate Classes (Rubel et al., 2017; Kottek et al., 2006) and WRB and USDA soil taxonomy systems derived from SoilGrids (Hengl et al., 2017). Please refer to lines 333 to 341 and figure 3 to 5.

Lines 333-341: Fig. 3 shows the number of samples by climatic zones (Rubel et al., 2017; Kottek et al., 2006). Majority of the data is from warm temperate, fully humid climate (49%), arid steppe climate and warm temperate climate with dry summer are the second and third most represented climate classes with 22 and 12 % respectively. On the other hand, Fig. 4 and 5 show the frequency of experimental sites respectively by WRB and USDA soil taxonomy systems based on the SoilGrids dataset (Hengl et al., 2017). Regarding the WRB classification system (Fig. 4), in total, 35 WRB reference soil subgroups are included among experimental sites where 55% of the experimental sites comprised four subgroup classes of Haplic Acrisols (8%), Haplic Luvisols (11%), Haplic Calcisols (15%), and Haplic Cambisols (21%). While 29 soil suborders classes of USDA soil taxonomy are included in this study (Fig. 5) where Udalfs (9%), Orthents (9%), and Ustolls (9%) are the most frequently appeared soil suborders in this investigation.

Figure 3- Number of samples by Köppen-Geiger climatic zones (Rubel et al., 2017; Kottek et al., 2006)

Figure 4- Frequency of WRB reference soil subgroups in experimental sites derived from SoilGrids (Hengl et al., 2017)

Figure 5- Frequency of USDA soil suborders in experimental sites (Hengl et al., 2017)

Comment: 3. A set of ancillary variables are introduced in the data set. The intention

of this is to apply data mining techniques to explain the infiltration process parameters and hence to allow developing new explanatory or predictive models. However, the quality of these models will depend on the quality of the ancillary variables that have been introduced in the database. Unfortunately, some ancillary variables are proposed in Table 4 that are not well defined or not well normalized or standardized. The added value of adding these ancillary variables to the database should be reconsidered. This is particularly the case for FC (many definitions of the field water capacity can be retrieved in the literature, see e.g. Nachabe, 1998), soil pH (measured in water or in KCl), and wet aggregate stability.

Response: Thank you to highlight this important concern. Providing detailed information on each soil properties is beyond the scope of the present manuscript. However, we have supplied the reference for all data (if available) that people can check the methodologies if needed. In case of those soil properties which we used for the presented analysis we simply assumed that measurement methodology did not significantly influenced the assumptions. For further use of the dataset harmonization will be indispensable e.g. generation of pedotransfer functions. In lines 288-290 and 496-506 we highlight it for the readers. Some variables were detailed because they are frequently part of the soil descriptive indicators (as pH, FC) concerning other scientific communities (i.e., geochemistry, agriculture). We preferred to keep these, in case SWIG database would interest sciences from other areas of expertise. Note that for PCA, statistical analyses we use only variables expected to play a role on a physical ground.

Lines 288-290: The references and correspondences for data supplied by direct communications with researchers are also reported in Table 2. Therefore, users may refer to these references for detailed information about the applied methods or procedures.

Lines 496-506: With respect to the transcription error, a strong effort has been made to double check data transcription to prevent or at least to minimize any probable error of this nature. Values of soil properties such as textural composition are known to

vary strongly between different laboratories labs and measurement methods. This is especially true for the finer textural classes like clay. Unfortunately, information on the measurement used to determine soil properties is most of the time lacking or insufficient to assess the magnitude of errors or biases. Internationally, there are typically more than one standard method to measure soil properties and several methods may have been applied to measure the reported soil characteristics. In this regard, no conversion has been made and only raw data are reported in database. However, we have supplied the reference for all data (if available) that people can check the methodologies if needed. Although supplying such information for each soil property may facilitate the use of database, but it will need a lot of additional work that could not be performed at this stage of development of the database. Such a work could be the purpose of a second version of the database that any reader should feel free to undertake to do.

Comment: 4. A very limited and preliminary data exploration analysis by means of PCA is presented in section 3.6. The preliminary and limited scope of this analysis questions the overall results of this analysis. For instance, the sorptivity, S, has been integrated into the PCA analysis. Yet S is not an intrinsic time-invariant soil property, but a soil variable that strongly is affected by the initial and saturated water content. Mixing such time dynamic state variable with static properties (such as soil texture, Ks, : : :) in a PCA has little sense, as the results will strongly depend on the initial water content before the infiltration will start.

Response: The authors thank the reviewer for these precisions. Sorptivity, S, and hydraulic conductivity, Ks were analyzed since they are two parameters that define water infiltration, considering the studied equations used for modelling water infiltration (e.g., equations 6-7). We agree with the reviewer that, in opposite to Ksat, S is not an intrinsic permeable for the soil. S depends upon the soil hydraulic function and initial and final water contents. In this study, we decided to include S in the PCA for completeness and not with the aim of seeking correlations. Interestingly it shows a small but positive correlation with Ksat with deserve further attention as these are two

quantities to be estimated from infiltration data.

Comment: Line 170. "In addition to its global coverage". Cf. above. Global coverage should be demonstrated by a representatively analysis.

Response: The sentence was slightly changed to avoid an over globalization of the data.

Comment: Line 176. We should expect that land use can be assessed for all the cases. If the spatial coordinates of the infiltration data are known, land use can be retrieved from historical land use data archives (see Google Earth Engine).

Response: Yes, the reviewer is right. However, this is out of the scope of this activity. It is also questionable whether the retrieved land use will correspond to the actual land use of a point scale measurement given the discrepancies in spatial scales. We have reported the land use when it was available but adding data without on-site verification is not a correct way to proceed. This is what we can consider in a next generation of SWIG when we will have more time to evaluate the accuracy of the reported land uses and perform on-site verifications.

Comment: Line 199. "In general, the soil infiltration rate decreases nonlinearly over time". Cf above. This is specifically the case when the infiltration process is studied under controlled conditions (typically as the cases where controlled infiltration experiments are performed). In general, infiltration is very time dynamic, conditioned to time variable climatic conditions, and in-situ infiltration rates will not 'in general' decrease with time. It is therefore suggested to give a clear definition of "infiltration" in this paper and make a clear distinction between in-situ and controlled experimental infiltration processes.

Response: The manuscript was revised accordingly, including several sentences to introduce in more details the concept of water infiltration and its measure on the field. Please refer to lines 199-216.

Lines 199-216: Two main mechanisms are responsible for the generation of excess water that produce overland flow: Dunne saturation excess and Hortonian infiltration excess (Sahoo et al., 2008). Dunne overland flow or saturation excess occurs when the soil profile is completely saturated and precipitation can no longer infiltrate into soil. The Dunne mechanism is more common to near-channel areas or it is generated from partial areas of the hillslope where water tables are shallowest (Sahoo et al., 2008). On the other hand, Hortonian overland flow is characterized by rainfall intensities exceeding the infiltration rate of the soil. In the other words, during a rainfall event, water infiltration at the soil surface and runoff are highly conditioned by the boundary condition, namely, under field condition, of the rainfall intensity and the soil hydraulic properties. If the rainfall intensity is lower than the soil infiltration rate, water will completely infiltrate into the soil without any runoff (Hillel, 2013). In this case, the infiltration rate will align with the rainfall intensity. Otherwise, if the precipitation intensity exceeds the soil infiltration rate at a certain moment in time, excess water will be generated even if the soil profile is unsaturated. In this case water will pond on the soil surface and becomes available for surface runoff. In this case, the boundary condition at soil surface shifts from imposed flow rate to imposed water pressure head. Admitting that the water pressure heads remain constant at the soil surface, the infiltration rate describes a decreasing function over time and tending towards the value of the hydraulic conductivity corresponding to the water pressure head at the surface (Angulo et al., 2016, Chow et al., 1988). In the past decades, water infiltration tests, using either ponded or tension infiltrometers have been developed to quantify the cumulative infiltration at the soil surface. In this case, the 3D axisymmetric water infiltration corresponds to an upper boundary defined by a constant water pressure head or a series of constant water pressure heads.

Comment: Line 207. "However, as infiltration proceeds, the gradient: : :." This is only the case when a pressure head boundary condition is used to define the infiltration process. This may not be the case when flux boundary conditions are used (e.g. constant precipitation). For instance, in case infiltration in a dry soil is analyzed subjected to constant flux boundary conditions, with an imposed flux that is smaller than the saturated hydraulic conductivity of the soil, than no ponding will occur, all water will infiltrate, and no decrease of pressure gradients will be observed.

Response: See comment above.

Comment: Line 209. ": : :approximates saturated hydraulic conductivity". This definition is often debated in the literature. For instance, Kutilek and Nielsen (1994), suggest Ks = 2/3 * the asymptotic value.

Response: When water infiltration experiments are performed, under constant water pressure head at surface, 1D infiltration rate decreases from infinity to the final infiltration rate that corresponds to gravity flow, i.e. the value of the hydraulic conductivity at the imposed water pressure head. Would it be zero, the infiltration rate will tend towards the value of the saturated hydraulic conductivity (see for instance Haverkamp et al., 1994). For some soils (in particular fine soils with very low permeability), steady state is unreachable within reasonable times. For these cases, two term equations (Vandervaere et al., 2000) can be used to describe cumulative infiltration and infiltration rate. In such a case, the final slope can be linear functions of Ks, for which Kutilek and Nielsen proposed 2/3.

Comment: Line 218. The Richards equation written in water content form is often referred to as the Fokker-Planck water diffusion equation.

Response: Done

Comment: Line 245. ".. from all over the globe". Be more rigorous. Many parts of the globe have not been considered for data collection.

Response: Done

Comment: Line 278. To avoid confusion, define Mi exactly.

Response: The variable was change to D, for particle diameter. Indeed, as stated in the

revised version, Di corresponds to the geometric average of interval limits that define three main fractions of sand, silt, and clay with mean values of 0.001, 0.026, and 1.025 mm, respectively.

Comment: Line 302. Please reformulate this phrase (what is the principle phrase?).

Response: Done

Comment: Line 327. This statement is clearly not supported by the data in Figure 4. Please avoid general statements that are not supported by the data, or introduce cautionary notes to put such statements in a correct perspective.

Response: The previous figures 3 and 4 were removed in the revised manuscript. We changed the sentence. Land use is known to impact soil structure and thus water infiltration processes. We then change the sentence to turn it into a general statement and add a reference.

Comment: Line 363. Correct: "Matlab$^{TM}$".

Response: Done

Comment: Line 372. The "material and methods" section does not explain in detail the difference between those two approaches. What is exactly meant by 'measured Ksat'?

Response: The measured values of Ksat were obtained by other means by the contributors and tabulated in SWIG database. The comparison of the orders of magnitude of the values obtained by fitting water infiltration data, referred to as estimated Ks with the values tabulated in SWIG database (rfred to as measured one) and the previous studies for the different classes of soil helps to validate the estimated values. This is already described in lines 424-426.

Lines 424-426: The measured values of Ksat were obtained by other means by the contributors and tabulated in SWIG database. Reefer to reference for collected data for detailed information of applied means to measure Ksat.

Comment: Line 888. If Table 3 is the continuation of Table 2, then it should not be a new table. (So no increase in table number).

Response: Table 3 was turned into Table 2.

Comment: Line 911. For improving the comparability, please harmonized the data in the same units (eg. log values in cm/day).

Response: It is true that for the comparability harmonization would give a much clearer view, but full dataset were not available to derive mean values uniformly in log10 cm/day

Comment: Line 918. Figure 3 adds very little information to the manuscript and is not very useful for the reader. Please consider to eliminate.

Response: Done

Please also note the supplement to this comment:
https://www.earth-syst-sci-data-discuss.net/essd-2018-11/essd-2018-11-AC2-supplement.zip